# LLM-based Symbolic Regression with Tool-Augmented Multi-Objective Optimization

## Abstract

Symbolic Regression (SR) aims to discover analytical equations from observational data and plays a central role in scientific modeling. While recent Large Language Model (LLM) based approaches show promise, they face two key limitations. First, they lack dedicated data analysis mechanisms to uncover variable dependencies, which reduces the efficiency of equation discovery. Second, most methods rely on single-objective evaluation focused solely on fitting error. This neglect of structural complexity and generalization often causes models to converge prematurely to local optima, limiting their ability to explore the broader equation space. To address these issues, we propose Tool-Augmented Multi-Objective Symbolic Regression (TAMOSR), a unified framework that integrates external analytical tools (e.g., correlation analysis, mutual information, periodicity detection) to extract structural priors and guide equation generation, while simultaneously optimizing for accuracy, complexity, and generalization via a multi-objective evaluation module with a dynamic Pareto front. TAMOSR employs two collaborative LLM modules: a Meta Strategy Generator, which selects tools and synthesizes structural optimization strategies based on Pareto-optimal equations, and an Equation Generator, which produces new candidate equations accordingly. The system operates in a closed loop, continuously refining both strategies and equation structures. Experiments on diverse benchmarks demonstrate that TAMOSR outperforms existing SR methods in accuracy, generalization, and search efficiency, offering a scalable and adaptable paradigm for scientific discovery.

## 1 Introduction

Symbolic Regression (SR) (Makke & Chawla, 2024b) aims to discover underlying mathematical equations from data and has long been recognized as a key methodology in scientific discovery. It has been widely applied across disciplines, from identifying physical laws (Makke & Chawla, 2024a; Reuter et al., 2023) and modeling chemical systems (Chen et al., 2025; Deng et al., 2023), to analyzing dynamic processes in biological or economic systems (Wahlquist et al., 2024; Shi et al., 2024). By generating compact and interpretable equations, SR enables both accurate prediction and deep insight into system behavior.

SR has long been recognized as an NP-hard problem (Virgolin & Pissis, 2022), motivating diverse algorithmic developments. Early approaches based on genetic programming (Schmidt & Lipson, 2009; Cranmer, 2023) evolve equations via mutation and crossover. Reinforcement learning (Petersen et al., 2021) models SR as a sequential decision-making process. Recently, Transformer-based models have enabled end-to-end learning from data to equations (Biggio et al., 2021; Kamienny et al., 2022; Zhang et al., 2025). With the rise of large language models (LLMs), methods such as LLM-SR (Shojaee et al., 2025a) and LaSR (Grayeli et al., 2024) leverage LLM's in-context learning capabilities and scientific priors to perform symbolic reasoning and equation generation.

Despite encouraging progress, existing LLM-based SR methods face two key limitations. First, they typically lack systematic analysis of variable dependencies and data distributions, relying instead on problem descriptions as context. This often results in poorly constrained search spaces, which limits both the efficiency and directionality of equation exploration. Second, most approaches adopt a single-objective evaluation, typically minimizing fitting error, while overlooking other critical

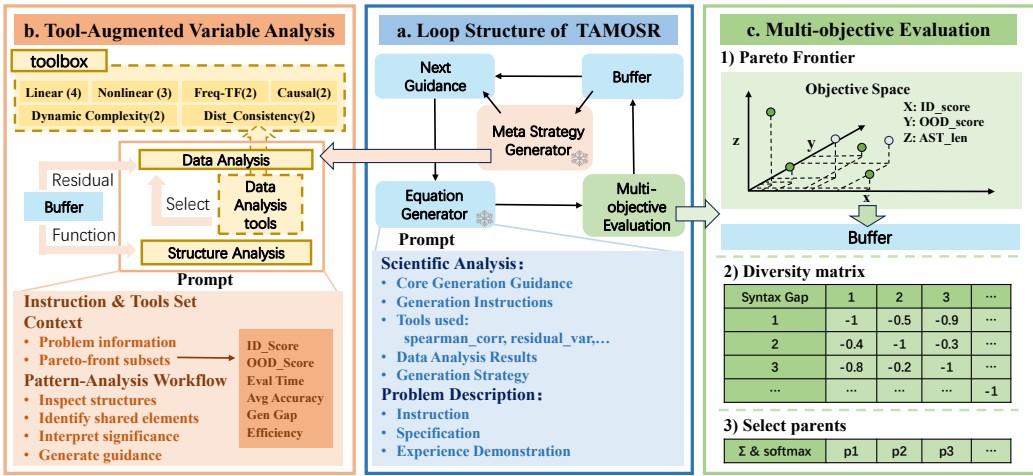

Figure 1: Overview of TAMOSR. (a) TAMOSR integrates a Meta Strategy Generator and multi-objective evaluation into the SR cycle. (b) The Meta Strategy Generator receives candidate equations and their residual statistics from the buffer, performs structural analysis, and selects appropriate tools for data-driven variable analysis. (c) The multi-objective evaluator assesses equations along three dimensions, and selects new parents via a diversity-guided sampling process.

factors such as equation complexity, generalization, and diversity. This can lead to overfitting and premature convergence to locally optimal solutions.

To address these challenges, we propose Tool-Augmented Multi-Objective Symbolic Regression (TAMOSR), a framework (Figure 1(a)) inspired by the human scientific modeling process. Scientists typically begin by analyzing data using a variety of tools, evaluate candidate hypotheses from multiple perspectives, and iteratively refine their modeling direction accordingly. TAMOSR operationalizes this human-like modeling paradigm through two core mechanisms and a cooperative, LLM-driven evolution system.

At its core, TAMOSR first employs a tool-augmented analysis mechanism that invokes a suite of analytical tools to extract variable relationships from multiple complementary dimensions, such as linearity, periodicity, and causality. These insights are converted into interpretable priors and structural constraints that guide equation generation. In parallel, a multi-objective evaluation mechanism (Figure 1(c)) jointly assesses candidate equations across three dimensions: accuracy on in-domain data, generalization to out-of-domain data, and structural complexity measured via abstract syntax tree (AST) length. A Pareto front is maintained using non-dominated sorting to preserve high-quality equations that represent optimal trade-offs.

To realize these mechanisms in an adaptive and iterative manner, TAMOSR incorporates two co-operating LLMs. The Meta Strategy Generator (Figure 1(b)) serves as a high-level planner: it synthesizes insights from tool outputs and structural patterns from the Pareto front to create natural language refinement strategies. The Equation Generator (Figure 1(a)) consumes these strategies to generate structurally novel and performance-driven candidates. Together, these components form a self-improving feedback loop between analytical insight and symbolic structure evolution.

We evaluate TAMOSR on five benchmarks spanning physics, chemistry, biology, and materials science, using both the open-source LLaMA-3.1 (Kassianik et al., 2025) and the commercial GPT-4o mini (OpenAI, 2024). Across all tasks, TAMOSR consistently outperforms traditional SR methods and recent LLM-based baselines in terms of accuracy, generalization, and equation compactness.

## 2 PRELIMINARIES

In SR, the learning task typically starts with a dataset consisting of input-output pairs:

$$D = \{(\mathbf{x}_i, y_i)\}_{i=1}^n, \quad \mathbf{x}_i \in \mathbb{R}^d, \quad y_i \in \mathbb{R},$$

where $\mathbf{x}_i$ denotes a $d$-dimensional input vector and $y_i$ is the corresponding scalar output. The goal is to discover an explicit analytical equation $f(\cdot)$ such that the predicted outputs $\hat{y}_i = f(\mathbf{x}_i)$ closely approximate the ground truth $y_i$. To assess the quality of a candidate equation, the Normalized Mean Squared Error (NMSE) is often used:

$$\text{NMSE}(f, D) = \frac{1}{n} \sum_{i=1}^{n} \left( \frac{f(\mathbf{x}_i) - y_i}{\sigma_y} \right)^2,$$

where $\sigma_y$ is the standard deviation of the target values across the dataset $D$. This metric reflects the equation's predictive accuracy, normalized by the variance of the outputs. Beyond fitting accuracy, SR also values simplicity and generalization, seeking equations that are not only accurate but also compact and transferable to unseen domains.

Our work builds on LLM-SR, a framework that leverages LLMs to generate symbolic equations through iterative optimization. Its core pipeline includes:

- Equation skeleton generation: Structured prompts contain task-specific information (e.g., variable names, optimization goals, example equations), guiding the LLM to produce physically plausible equation skeletons.
- Parameter optimization and scoring: The skeletons' free parameters are optimized (e.g., via BFGS (Fletcher, 1987) ) and scored using mean squared error.
- Experience buffer and feedback: High-quality equations are retained and reused as in-context examples, enabling iterative refinement through feedback-driven generation.

While promising, current LLM-based SR methods faces key limitations: they lack systematic modeling of variable dependencies, leading to structurally under-informed equations; and they rely on a single-objective evaluation focused solely on fitting error, neglecting factors such as complexity and generalization. These shortcomings diminish their effectiveness on solving more challenging tasks.

## 3 METHOD

To address the above limitations, we propose **TAMOSR** (Tool-Augmented Multi-Objective Symbolic Regression), a unified framework that integrates external data analysis tools, multi-objective evaluation, and cooperative LLMs to enhance equation quality and search efficiency. TAMOSR first extracts structural priors by analyzing variable relationships with diverse analytical tools. It then introduces a multi-objective evaluation mechanism that jointly considers fitting error, equation complexity, and generalization, dynamically maintaining a Pareto front through non-dominated sorting. Finally, two complementary LLMs work in tandem: one generates structural refinement strategies, while the other synthesizes candidate equations accordingly, forming a closed-loop process that continuously guides and improves equation discovery. We next detail TAMOSR's core components.

### 3.1 TOOL-AUGMENTED VARIABLE ANALYSIS

This component constructs structural priors by quantifying diverse variable relationships using a suite of carefully designed analytical tools. Details of these tools are provided in Appendix G.

**Linear correlation tools** are essential for identifying dominant variables and constructing interpretable regression structures. TAMOSR integrates several complementary methods to assess linear dependencies from multiple statistical angles. The *Pearson correlation coefficient* (Pearson, 1895) quantifies pairwise linear associations, especially effective for Gaussian-like data. *Simple linear regression* and *residual variance analysis* (Montgomery et al., 2013) evaluate predictive capacity and error stability. *PCA-based explained variance* (Jolliffe, 2002) identifies the key directions of structural variance. These tools jointly establish a solid basis for linear trend detection.

**Nonlinear dependency tools** are employed to capture complex interactions essential for modeling nonlinear systems. TAMOSR integrates three complementary methods: the *Spearman rank correlation* (Spearman, 1904) measures monotonic associations based on rank, offering robustness to noise and non-Gaussian data; *mutual information* (Shannon, 1948) measures the overall statistical dependency between variables without assuming any parametric form; and *mutual information regression*

(Pedregosa et al., 2011) quantifies the marginal contribution of each variable conditioned on others. Together, these tools guide whether nonlinear or higher-order terms should be introduced.

**Time-frequency analysis tools** help detect periodicity and transient dynamics, which frequently occur in oscillatory and multiscale systems. TAMOSR employs two complementary methods: *Fast Fourier Transform* (Cooley & Tukey, 1965) identifies dominant global frequency components, while the *wavelet transform energy spectrum* (Daubechies, 1992) captures localized, non-stationary fluctuations. These insights support the inclusion of periodic terms (e.g., $\sin$) in candidate equations.

**Causal inference tools** are used to identify whether one variable may influence another in a predictive or explanatory sense. TAMOSR adopts *Granger causality* (Granger, 1969) for detecting temporal causal influence in linear time-series, and *Convergent Cross Mapping* (Sugihara et al., 2012) for identifying latent causality in nonlinear systems with potential delays. These methods provide structural signals that enhance the interpretability and explanatory power of generated equations.

**Dynamic complexity tools** help assess intrinsic system richness and redundancy, guiding the pruning of over-specified components in candidate equations. TAMOSR employs three representative methods: the *Lyapunov exponent* (Wolf et al., 1985), which measures sensitivity to initial conditions and indicates chaotic behavior; the *correlation dimension* (Grassberger & Procaccia, 1983), which estimates the system's effective degrees of freedom; and *Dynamic Time Warping (DTW)* (Sakoe & Chiba, 1978), which evaluates time-shifted similarity between variable trajectories. Together, they offer structural cues for constructing compact and robust equations.

**Distribution consistency tools** evaluate whether input variables behave uniformly across different input regions. TAMOSR uses the *Kolmogorov–Smirnov (KS) test* (Massey, 1951) to detect distributional shifts between subdomains, informing the use of region-dependent structures to reflect local variations in the data distribution.

Rather than introducing new tools, the key innovation of TAMOSR lies in enabling LLMs to autonomously invoke and coordinate these tools to extract core dependency patterns among variables. These insights are distilled into concise guidance that informs variable selection and function composition, thereby improving responsiveness to data characteristics and enhancing the scientific plausibility of generated equations. By adaptively combining outputs from heterogeneous analyses, for instance by linking correlation measures with periodicity detection, TAMOSR supports more targeted and reliable equation discovery. The impact of this tool-augmented analysis on equation generation, is substantiated by the case studies in Appendix H.3. Looking ahead, TAMOSR also opens the possibility for LLMs to synthesize new tools, further expanding the scope of SR research.

### 3.2 Multi-Objective Evaluation

To enhance search efficiency and model quality, TAMOSR adopts a multi-objective evaluation scheme that jointly considers predictive accuracy, generalization, and structural simplicity. A Pareto-based selection strategy maintains a diverse set of non-dominated candidate equations, improving robustness and exploration of the solution space.

#### Evaluation Metrics

Different from prior LLM-based SR methods that optimize only fitting error, TAMOSR employs a three-fold metric framework:

**(1) In-Domain (ID) Accuracy.** Measures interpolation performance within the central region of the input space. The in-domain subset $D_{\text{ID}}$ is selected from the middle intervals along each input dimension. As shown by the shaded regions in Figure 2, these correspond to percentile-based ID bounds. Accuracy is quantified using NMSE, denoted as $\text{NMSE}_{\text{ID}}$.

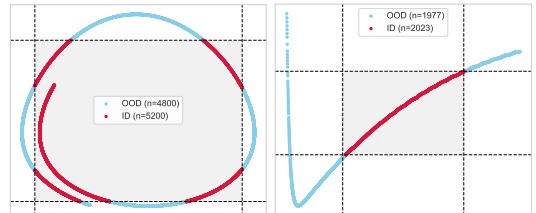

Figure 2: Illustration of the separation between ID and OOD samples across two example problems.

**(2) Out-of-Domain (OOD) Generalization.** Measures extrapolation performance in the peripheral regions $D_{\text{OOD}}$, which are formed by excluding the central intervals. These correspond to the white

regions in Figure 2. NMSE on this subset ($\text{NMSE}_{\text{OOD}}$) reflects the model's robustness to distributional shifts, a key requirement for scientific applications.

**(3) Equation Complexity.** To promote interpretability, equation complexity is measured via the size of its abstract syntax tree (AST) (Neamtiu et al., 2005). In TAMOSR, each equation is represented as a Python function and parsed into an AST, where each node corresponds to a variable, operator, or function call. The total node count provides a language-independent and semantically meaningful estimate of structural complexity, guiding the search toward compact equations.

PARETO-GUIDED MULTI-OBJECTIVE OPTIMIZATION

Based on the above three metrics, TAMOSR formulates equation discovery as a multi-objective optimization problem over a candidate set $\mathcal{P} = \{f_1, f_2, \ldots, f_M\}$, aiming to balance fitting accuracy, extrapolation, and structural simplicity:

$$f^* = \arg\min_{f \in \mathcal{P}} \left( \text{NMSE}_{\text{ID}}(f),\ \text{NMSE}_{\text{OOD}}(f),\ \text{ASTLen}(f) \right)$$

A function $f_a$ dominates $f_b$ if it performs no worse across all objectives and better on at least one. The Pareto front $\mathcal{P}^*$ consists of all non-dominated candidates in $\mathcal{P}$. To improve the quality of the front, TAMOSR filters out overly simple candidates with low equation complexity that may mislead the search by applying lower-bound thresholds on both ID and OOD NMSE. Remaining equations are ranked via non-dominated sorting to construct the current Pareto-optimal set.

## 3.3 COOPERATIVE LLMS FOR EQUATION EVOLUTION

To integrate variable-aware analysis with multi-objective optimization, TAMOSR employs a cooperative framework consisting of two LLMs: the *Meta Strategy Generator* $\pi_{\text{stg}}$ and the *Equation Generator* $\pi_{\text{eq}}$. The former extracts structural search strategies, while the latter generates candidate equations accordingly, jointly driving the population toward Pareto-optimality (Algorithm 1).

**Meta Strategy Generator.** $\pi_{\text{stg}}$ formulates equation search strategies by synthesizing variable-level insights and structural abstraction. At each iteration, given the current Pareto front $\mathcal{P}^*_{t-1}$, $\pi_{\text{stg}}$ autonomously selects a subset of tools $\mathcal{A} \subseteq \texttt{Toolbox}$, applies them to the dataset $\mathcal{D}$, and summarizes key relationships as natural language descriptions $\mathcal{R}_{\text{varrel}}$ to guide downstream generation.

In parallel, $\pi_{\text{stg}}$ performs structural abstraction over $\mathcal{P}^*_{t-1}$, producing complementary guidance $\mathcal{R}_{\text{struct}}$ through: 1) *Commonality Extraction*, which identifies frequent substructures that define prevailing symbolic motifs; 2) *Disparity Analysis*, which diagnoses regional residual patterns to uncover structural weaknesses; and 3) *Blind Spot Discovery*, which detects unexplored symbolic components to encourage diversity. The final strategy $\mathcal{S}_t = (\mathcal{R}_{\text{varrel}}, \mathcal{R}_{\text{struct}})$ is passed to the equation generator $\pi_{\text{eq}}$ to steer the next round of generation.

**Equation Generator.** Upon receiving the strategy $\mathcal{S}_t$ from $\pi_{\text{stg}}$, $\pi_{\text{eq}}$ is responsible for synthesizing a new batch of candidate equations $\mathcal{P}_t$. To promote structural diversity and prevent premature convergence, TAMOSR incorporates a *structure-diversity–guided parent sampling module*, which explicitly prioritizes structurally distinct candidates during equation generation.

Specifically, each equation $f \in \mathcal{P}^*_t$ on the Pareto front is parsed into an abstract syntax tree (AST), from which a set of symbolic subtrees $S(f)$ is extracted. The pairwise structural dissimilarity between two equations is computed via a subtree-overlap metric defined as:

$$\text{SyntaxDiv}(f_i, f_j) = -\frac{|S(f_i) \cap S(f_j)|}{|S(f_i)|}.$$

This score reflects the relative uniqueness of $f_i$'s structure compared to $f_j$. For each equation, we compute the average structural diversity score:

$$\text{Score}_{\text{div}}(f_i) = \tfrac{1}{N-1} \sum_{j \neq i} \text{SyntaxDiv}(f_i, f_j).$$

A softmax sampling procedure is then applied to construct a parent set $\mathcal{P}_{\text{parent}}$, biased toward structurally diverse equations. These parents serve as in-context examples, which, together with the

strategy $\mathcal{S}_t$, are fed into $\pi_{\text{eq}}$ to generate new equation candidates: $f' \sim \pi_{\text{eq}}(\mathcal{S}_t, \mathcal{P}_{\text{parent}})$. This procedure facilitates the generation of structurally diverse and generalizable expressions, driving the continuous evolution and expansion of the equation population $\mathcal{P}$.

## 4 EXPERIMENTS

### 4.1 DATASETS

To assess TAMOSR's performance, we adopt two sets of challenging datasets. The first includes four standard benchmarks from LLM-SR, spanning nonlinear oscillatory systems (**Oscillation 1 & 2**), where Oscillation 1 focuses on periodic signal composition and Oscillation 2 introduces cross-variable interactions and non-periodic disturbances to increase modeling difficulty, an **E. coli Growth** task modeling multivariate biological dynamics with nonlinear couplings (Monod, 1949; Rosso et al., 1995), and a **Stress-Strain** task from materials science featuring piecewise nonlinear deformation behavior (Aakash et al., 2019).

The second dataset is **LSR-Synth–Chemistry** from LLM-SRBENCH (Shojaee et al., 2025b), which consists of 36 tasks derived from a chemical kinetics base equation with progressively increasing symbolic complexity. It is specifically designed to evaluate a model's ability to generalize across nested, unseen, and semantically rich expressions. In this work, we focus on LSR-Synth–Chemistry rather than the complete LLM-SRBENCH suite because the other three datasets are constructed variants of the four benchmark problems already included in our evaluation. By concentrating on LSR-Synth–Chemistry, which is both novel and complementary, we ensure a comprehensive yet non-redundant assessment of model performance while also taking into account the practical constraints of available hardware resources. Full dataset descriptions are provided in Appendix D.

### 4.2 BASELINES

We compare TAMOSR against a range of representative baselines from both classical and LLM-based SR methods. For the four standard tasks in the LLM-SR benchmark, we include GPLEARN, a classical genetic programming-based SR method; PYSR (Grayeli et al., 2024), which combines evolutionary search with symbolic compression; UDSR (Landajuela et al., 2022), which replaces DSR's RNN policy with a pretrained Transformer and neural-guided decoding; RAG-SR (Zhang et al., 2025), which augments equation generation with structure retrieval; and LLM-SR (Shojaee et al., 2025a). On the more challenging LSR-Synth–Chemistry, we compare TAMOSR with leading LLM-enhanced methods, including SGA (Ma et al., 2024), which combines LLM-based hypothesis generation with physics-informed parameter optimization via bilevel search, and LASR (Grayeli et al., 2024), which extracts abstract symbolic concepts from prior equations to guide hybrid LLM-evolutionary equation generation.

### 4.3 EVALUATION METRICS

We evaluate different methods using three metrics: (1) Accuracy to tolerance $\tau$, denoted as $\text{Acc}_{\text{all}}(\tau)$ and $\text{Acc}_{\text{avg}}(\tau)$, and (2) Normalized Mean Squared Error (NMSE). $\text{Acc}_{\text{all}}(\tau)$ measures task-level correctness by requiring all test points to satisfy the relative error bound $\tau$, i.e., $\text{Acc}_{\text{all}}(\tau) = \mathbf{1}\left(\max_{1 \leq i \leq N_{\text{test}}} \left|\frac{\hat{y}_i - y_i}{y_i}\right| \leq \tau\right)$. $\text{Acc}_{\text{avg}}(\tau)$ computes the proportion of test points that meet the same criterion, defined as $\text{Acc}_{\text{avg}}(\tau) = \frac{1}{N_{\text{test}}} \sum_{i=1}^{N_{\text{test}}} \mathbf{1}\left(\left|\frac{\hat{y}_i - y_i}{y_i}\right| \leq \tau\right)$.

We further evaluate the quality of the final Pareto front using Hypervolume (HV) (Zitzler & Thiele, 1999) and Inverted Generational Distance (IGD) (Zitzler et al., 2000), two standard indicators in multi-objective optimization (see Appendix F for details). HV measures the volume dominated by the obtained solution set with respect to a fixed reference point, capturing both convergence and diversity. It reflects the overall coverage of the objective space and favors solution sets that are both well-converged and diverse. IGD computes the average distance from the ground truth equations to its nearest counterpart in the generated front, emphasizing approximation accuracy with respect to the true Pareto-optimal equations. For LLM-SR, we construct a pseudo Pareto front by collecting all nondominated solutions discovered across 100 generations to enable fair comparison.

| Model | Oscillation 1 | | Oscillation 2 | | E. coli growth | | Stress-Strain | |
|---|---|---|---|---|---|---|---|---|
| | $Acc_{avg\text{-}0.001}(\%)\uparrow$ | NMSE↓ | $Acc_{avg\text{-}0.001}(\%)\uparrow$ | NMSE↓ | $Acc_{avg\text{-}0.1}(\%)\uparrow$ | NMSE↓ | $Acc_{avg\text{-}0.1}(\%)\uparrow$ | NMSE↓ |
| GPlern | 0.11 | 0.0972 | 0.05 | 0.2000 | 0.76 | 1.0023 | 28.43 | 0.3496 |
| PySR | 3.80 | 0.0003 | 7.02 | 0.0002 | 2.80 | 0.4068 | 70.60 | 0.0347 |
| RAG-SR | 39.47 | 1.49e-6 | 0.43 | 0.0282 | 2.04 | 0.2754 | 76.28 | 0.0282 |
| uDSR | 1.78 | 0.0002 | 0.36 | 0.0856 | 1.12 | 0.5059 | 59.15 | 0.0639 |
| LaSR (Llama-3.1) | 2.79 | 0.7485 | 1.09 | 0.0310 | 3.44 | 0.1349 | 71.84 | 0.0320 |
| LLM-SR (Llama-3.1) | 12.67 | 2.55e-5 | 8.20 | 4.70e-5 | 1.36 | 0.5815 | 76.21 | 0.0333 |
| LLM-SR (4o-mini) | 11.12 | 2.07e-5 | 8.66 | 4.51e-5 | 3.24 | 0.0863 | 71.28 | 0.0491 |
| TAMOSR (Llama-3.1) | **100.00** | **1.27e-15** | 99.45 | **1.70e-10** | **6.60** | 0.0208 | 85.02 | 0.0150 |
| TAMOSR (4o-mini) | 99.99 | 1.42e-13 | **99.57** | 4.25e-10 | 6.32 | **0.0178** | **86.33** | **0.0144** |

Table 1: Overall performance of TAMOSR and baseline methods on four benchmarks.

## 4.4 TAMOSR CONFIGURATION

For fair comparison, we adopt the same LLMs across all methods: LLAMA-3.1-8B-INSTRUCT and GPT-4O-MINI, covering both lightweight and high-performance scenarios. In each iteration, TAMOSR generates four candidate equations for evaluation. The total number of iterations is set to 2000 for standard benchmarks and 1000 for LSR-Synth–Chemistry, following the LLM-SRBENCH protocol. Traditional baselines are allowed more iterations to ensure convergence. Additional details, including prompt design and sampling configurations, are provided in Appendix J and C.

## 5 FINDINGS

### 5.1 TAMOSR ACHIEVES THE BEST OVERALL PERFORMANCE

As shown in Table 1, TAMOSR consistently outperforms both classical and LLM-based SR baselines, achieving significantly lower NMSE and higher strict accuracy across all benchmarks. For example, on Oscillator 1 with LLaMA, TAMOSR reaches an NMSE of $1.27 \times 10^{-15}$, far surpassing LLM-SR's $2.55 \times 10^{-5}$. With both LLaMA and GPT-4o-mini, TAMOSR attains over 90% accuracy on several datasets. On the more challenging LSR-SYNTH–CHEMISTRY (Table 2), TAMOSR achieves the best overall performance with an average NMSE of $3.85 \times 10^{-7}$ and 86.1% accuracy, significantly outperforming other LLM-based models. These results validate the effectiveness of TAMOSR's tool-augmented multi-objective framework in enabling the discovery of more accurate equations.

| Model | $Acc_{all\text{-}0.1}(\%)\uparrow$ | NMSE↓ |
|---|---|---|
| SGA | 8.33 | 0.0458 |
| LaSR | 27.77 | 2.77e-04 |
| LLM-SR | 66.66 | 8.01e-06 |
| TAMOSR | **86.11** | **3.85e-07** |

Table 2: Comparison on LSR-Synth–Chemistry.

### 5.2 TAMOSR SHOWS STRONGER OOD GENERALIZATION

We evaluate TAMOSR's generalization ability under both ID and OOD across multiple benchmarks. As shown in Figure 3a, TAMOSR consistently achieves the lowest NMSE on all four standard tasks, significantly outperforming LLM-SR and other baselines. For instance, on the OOD split of Oscillator 1, TAMOSR reaches an NMSE of $6.20 \times 10^{-14}$, nearly eleven orders of magnitude lower than LLM-SR ($1.4 \times 10^{-3}$). Figure 3b further shows its superior median NMSEs across 36 chemistry tasks in LSR-Synth, under both ID and OOD conditions.

TAMOSR's generalization advantage stems from two key mechanisms. First, its multi-objective optimization explicitly incorporates OOD performance, guiding the search toward expressions that maintain low error on OOD inputs. Second, its meta-strategy module integrates variable-level analysis and structural diagnostics: the former identifies stable input dependencies via analytical tools, while the latter detects effective substructures and potential failure modes across the population. Together, these components help TAMOSR uncover symbolic relationships that generalize beyond distribution-specific patterns.

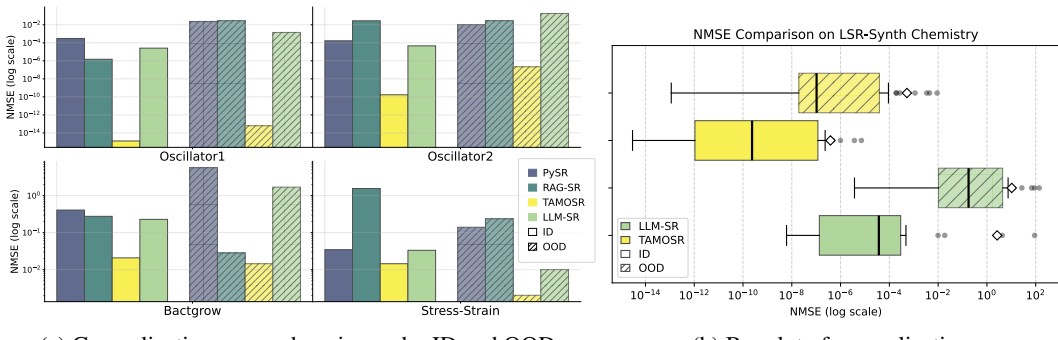

(a) Generalization across domains under ID and OOD.  (b) Boxplot of generalization.

Figure 3: Comparison of generalization across domains and chemistry task.

## 5.3 TAMOSR IMPROVES EQUATION DISCOVERY EFFICIENCY

We compare the convergence behavior of TAMOSR and LLM-SR across four benchmark tasks. As shown in Figure 4, TAMOSR not only reduces error more rapidly but also converges to lower final NMSE values. In most cases, it outperforms LLM-SR's best results (at 2000 iterations) within the first 1000 iterations. This efficiency gain stems from TAMOSR's meta-strategy design. By extracting key variable dependencies through analytical tools and summarizing structural patterns from the Pareto front, TAMOSR narrows the search space and avoids redundant exploration, steering the model toward high-quality equations with fewer iterations. These results highlight its efficiency in navigating the symbolic search space while maintaining both accuracy and generalization. We further provide the computational cost analysis in Appendix H.2.

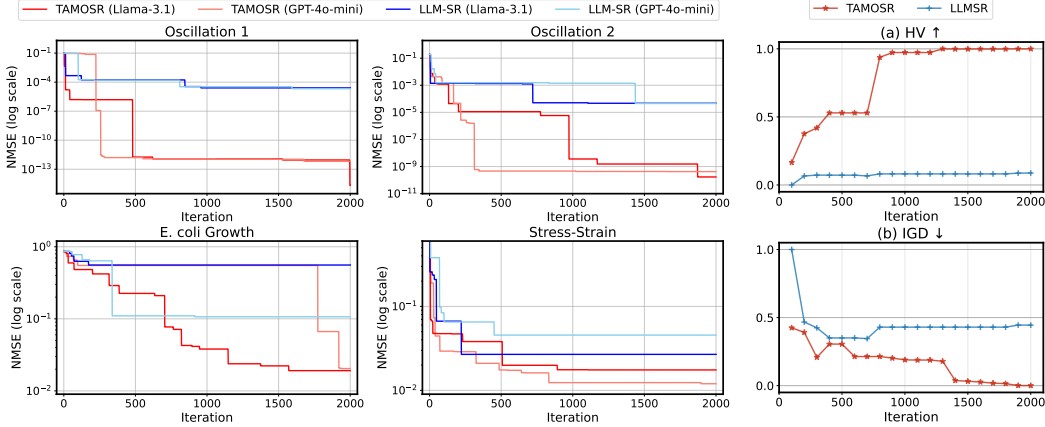

Figure 4: Training convergence curve comparison between TAMOSR and LLM-SR on four benchmarks.

Figure 5: Comparison of HV and IGD of different models.

## 5.4 TAMOSR EXHIBITS SUPERIOR MULTI-OBJECTIVE OPTIMIZATION QUALITY

We compare TAMOSR and LLM-SR on Oscillator 1 using HV and IGD. As shown in Figure 5a, TAMOSR achieves faster growth in HV compared to LLM-SR, indicating earlier and more effective discovery of diverse, high-quality non-dominated solutions. Similarly, Figure 5b shows that TAMOSR consistently reduces IGD at a faster rate, with better convergence toward the true Pareto front. LLM-SR, in contrast, plateaus earlier with higher IGD, reflecting a tendency to remain in suboptimal regions. These results demonstrate that TAMOSR significantly outperforms baselines in terms of convergence speed, solution diversity, and overall optimization quality.

### 5.5 ABLATION STUDY

To assess the contributions of key components in TAMOSR, we conduct ablation experiments on the Oscillator 1 benchmark with the LLaMA backbone. Our analysis focuses on two mechanisms: the multi-objective optimization mechanism and the meta strategy generator.

#### 5.5.1 IMPACT OF MULTI-OBJECTIVE OPTIMIZATION

We assess the impact of multi-objective optimization by replacing it with a single-objective variant (*w/o MultiObj*) that optimizes only fitting error, excluding complexity and generalization. In contrast, *w/o MultiObj* recovers only 1 of 9 symbolic terms, compared to 4 of 6 for TAMOSR:

**Ground Truth**

$$\dot{v} = 0.8\sin(x) - 0.5x \cdot v - 0.5v^3 - 0.2x^3 - x\cos(x).$$

**TAMOSR w/o MultiObj**

$$\dot{v} = a_0\tanh(a_1 x) - a_2\tanh(a_3 v) - a_4 x - a_5 v - a_6\ \boxed{xv} - a_7(x - v) - a_8(x + v) - a_9 v - (1 - a_9)x.$$

**TAMOSR**

$$\dot{v} = -a_0 x + a_1 v - a_2\ \boxed{x^3} + a_3\ \boxed{v^3} - a_4\ \boxed{xv} + a_5\ \boxed{\sin(a_6\,x)}\,.$$

This indicates multi-objective optimization is essential for discovering compact and interpretable equations. In contrast, single-objective optimization often leads to redundancy and overfitting.

#### 5.5.2 EFFECTIVENESS OF THE META STRATEGY GENERATOR

To further investigate the contribution of the meta strategy generator, we conduct a series of ablation studies by progressively removing its core components: the data analysis submodule (*w/o Data*), the structure analysis submodule (*w/o Struct*), and the entire module (*w/o Strategy*).

As shown in Figure 6, removing the data analysis module impairs strategy refinement based on variable relationships, leading to a notable performance drop. Excluding the structure module results in a noticeable decline in generalization and accuracy, as the model can no longer extract structural patterns from prior equations to guide generation. Removing the entire meta strategy generator results in prompt-only generation without feedback, producing the lowest accuracy and stability.

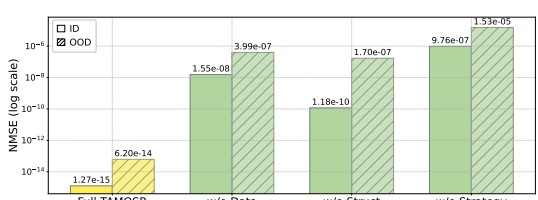

Figure 6: Ablation results on Oscillation 1.

These findings demonstrate that the data-driven and structure-guided feedback mechanisms offer complementary strengths in guiding TAMOSR's equation generation. Evaluations of the ablation settings on other benchmark datasets, are provided in Appendix E.1, and further ablation results on the role of different tools in the toolset are reported in Appendix E.2.

## 6 CONCLUSION, LIMITATION, AND FUTURE WORK

We introduced TAMOSR, a unified SR framework that integrates variable-level data analysis, multi-objective optimization, and cooperative LLMs to enhance the quality and efficiency of equation discovery. Through extensive benchmarks, TAMOSR consistently outperforms both classical and LLM-based baselines in terms of accuracy, generalization, and convergence, demonstrating its effectiveness in discovering interpretable and robust equations. TAMOSR currently focuses on a fixed set of evaluation objectives and relies on manually defined analytical tools. Future directions include expanding the objective space to incorporate domain-specific constraints, automating analysis tool discovery and refinement, and scaling to high-dimensional scientific datasets.

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

# APPENDIX

## A RELATED WORK

### A.1 SYMBOLIC REGRESSION

Traditional SR methods mainly rely on evolutionary algorithms, reinforcement learning (Petersen et al., 2021), and Transformers (Biggio* et al., 2021). For instance, genetic programming (Koza, 1990) formulates equation discovery as an evolutionary search over tree-based representations, refining structures via mutation and crossover. Reinforcement learning-based symbolic regression was first introduced by Petersen et al. (Petersen et al., 2021) and has since developed into various policy-optimization frameworks (Mundhenk et al., 2021; Landajuela et al., 2021; Crochepierre et al., 2022; Du et al., 2023). More recently, Transformer-based models (Valipour et al., 2021; Vastl et al., 2024; Kamienny et al., 2022; Li et al., 2023; Zhang et al., 2025) have been applied to SR, leveraging large-scale pretraining to improve equation generation. However, these models typically lack mechanisms to incorporate physical priors.

With advances in natural language processing, LLM-based SR methods such as LLM-SR (Shojaee et al., 2025a), LaSR (Grayeli et al., 2024), and ICSR (Merler et al., 2024) have emerged. LLM-SR utilizes scientific priors encoded in LLMs to generate plausible equation forms, followed by data-driven parameter fitting. LaSR introduces abstract concept generation to guide hypothesis construction, while ICSR formats training data as in-context prompts to induce function generation. Yet these approaches still rely heavily on pre-trained knowledge and lack explicit modeling of variable relationships or structured reasoning over data, which limits both their search efficiency and generalization ability.

### A.2 MULTI-OBJECTIVE OPTIMIZATION

Multi-objective optimization is a fundamental research direction in the optimization community and has been widely applied in areas such as combinatorial optimization (Lust & Teghem, 2012; Lin et al., 2022; Yao et al., 2025) and industrial design (Andersson, 2000; Cerda-Flores et al., 2022; Fromer & Coley, 2023). In the context of SR, Kommenda et al. systematically compared various complexity measures for multi-objective SR and proposed a new metric that preserves semantic information while improving search efficiency (Kommenda et al., 2015). Kubalík et al. introduced a physics-aware multi-objective approach (Kubalík et al., 2021) that jointly optimizes model accuracy and physical consistency, enhancing interpretability and reliability. TAMOSR is the first framework to integrate LLMs into multi-objective SR, enabling data-driven equation discovery with improved accuracy, simplicity, and generalization.

## B  ALGORITHMIC PSEUDOCODE FOR TAMOSR

---

**Algorithm 1** TAMOSR

---

1: **Input:** Training set $D$; Maximum iterations $T$; $n$ samples per iteration; Initial population $\mathcal{P}_0$ (optional); Meta Strategy Generator $\pi_{\text{stg}}$; Equation Generator $\pi_{\text{eq}}$; **Toolbox**: Data analysis toolset
2: **Output:** Approximate Pareto front $\mathcal{P}^*$
3: Initialize $\mathcal{P}_0$
4: **for** $t = 1, \ldots, T$ **do**
5:     $Res \leftarrow$ EvaluateEquationResiduals($\mathcal{P}^*_{t-1}, D$)
6:     $\mathcal{R}_{\text{varrel}} \leftarrow \pi_{\text{stg}}$.DataAnalysis($\textbf{Toolbox}, Res, \mathcal{P}^*_{t-1}$)
7:     $\mathcal{R}_{\text{struct}} \leftarrow \pi_{\text{stg}}$.GenerateStructuralPrompts($\mathcal{P}^*_{t-1}$)
8:     $\mathcal{S}_t \leftarrow (\mathcal{R}_{\text{varrel}}, \mathcal{R}_{\text{struct}})$
9:     $\mathcal{P}_{\text{parent}} \leftarrow$ ParentSelection($\mathcal{P}^*_{t-1}$)
10:    $\mathcal{P}_t \leftarrow \mathcal{P}^*_{t-1}$
11:    **for** $i = 1, \ldots, n$ **do**
12:        $f \sim \pi_{\text{eq}}(\mathcal{S}_t, \mathcal{P}_{\text{parent}})$
13:        $score \leftarrow$ MultiObjectiveEvaluation($f, D$)
14:        $\mathcal{P}_t \leftarrow \mathcal{P}_t \cup \{f, score\}$
15:    **end for**
16:    $\mathcal{P}^*_t \leftarrow$ PopulationManagement($\mathcal{P}_t$)
17: **end for**
18: $\mathcal{P}^* \leftarrow \mathcal{P}_T$

---

## C  TAMOSR CONFIGURATION AND LANGUAGE MODEL DETAILS

### C.1  TAMOSR CONFIGURATION

We implement TAMOSR with both open-source and commercial LLM backbones: **LLaMA-3.1-8B-Instruct** and **GPT-4o-mini**. The **LLaMA-3.1** model is quantized and deployed locally on NVIDIA H100 80GB GPUs for efficient inference. And **GPT-4o-mini** is accessed via the OpenAI API, providing high-quality reasoning without requiring local resources. Following the LLM-SRBENCH protocol, the maximum number of optimization iterations is set to 2000 for the four main benchmark datasets, and 1000 for the LSR-Synth–Chemistry dataset. In each iteration, the Meta Strategy Generator selects 3 tools from the tool set to analyze variable-level relationships and generates a natural language strategy prompt based on both data characteristics and symbolic structural patterns. Both the **Meta Strategy Generator** and **Equation Generator** in TAMOSR utilize LLM decoding with top-$k = 30$, top-$p = 0.3$, and temperature 0.6. In each iteration, the Equation Generator produces 4 candidate expressions. During Pareto frontier construction, we apply an NMSE-based filtering mechanism to eliminate trivial expressions: for each candidate, we compute the base-10 logarithm of its minimal NMSE (across ID and OOD), and discard it if it exceeds $10^{\lceil 0.5 \cdot \min \log_{10} \text{NMSE} \rceil}$, where the minimum is taken over the current candidate pool.

### C.2  LARGE LANGUAGE MODELS

TAMOSR employs two backbone LLMs: `LLaMA-3.1-8B-Instruct` and `GPT-4o-mini`, covering both open-source and API-accessible commercial models. The LLaMA-3.1 model is locally quantized to 4-bit precision and deployed on NVIDIA H100 80GB GPUs, requiring approximately 8GB of VRAM during inference, thus allowing execution on consumer-grade hardware with appropriate quantization. The GPT-4o-mini model is accessed via OpenAI's API, offering high-quality reasoning with minimal deployment overhead.

# D  DATASETS

## D.1  NONLINEAR OSCILLATOR

Oscillatory systems with nonlinear damping are foundational in physics and engineering for modeling the motion of objects subjected to restoring and dissipative forces. These systems are governed by second-order differential equations of the form $\ddot{x} + f(t, x, \dot{x}) = 0$, where the nonlinear function $f$ encapsulates dynamic interactions among position, velocity, and possibly time.

To evaluate a model's ability to recover such complex dynamics, two synthetic oscillator tasks are used. The first system is defined by:

$$\dot{v} = F \sin(\omega x) - \alpha v^3 - \beta x^3 - \gamma x v - x \cos(x) \quad (F = 0.8,\ \alpha = 0.5,\ \beta = 0.2,\ \gamma = 0.5,\ \omega = 1.0)$$

The second system follows:

$$\dot{v} = F \sin(\omega t) - \alpha v^3 - \beta x v - \delta x \exp(\gamma x) \quad (F = 0.3,\ \alpha = 0.5,\ \beta = 1.0,\ \delta = 5.0,\ \gamma = 0.5,\ \omega = 1.0)$$

with initial conditions $x = 0.5$, $v = \dot{x} = 0.5$, and simulation time $t \in [0, 50]$. These equations exhibit rich nonlinear structures and variable couplings, making them ideal benchmarks for testing symbolic reasoning and generalization beyond simple oscillatory behavior.

## D.2  BACTERIAL GROWTH

Accurately capturing the dynamics of E. coli proliferation under varying environmental conditions is of critical importance in areas such as biotechnology and microbiological risk assessment. This benchmark reflects realistic yet challenging biological modeling, where growth depends on multiple interacting factors.

The dataset models population change via:

$$\frac{dB}{dt} = f_B(B) \cdot f_S(S) \cdot f_T(T) \cdot f_{\text{pH}}(\text{pH})$$

where $B$ is bacterial density, $S$ is nutrient concentration, $T$ is temperature, and pH denotes acidity. Each term accounts for a distinct physiological influence on growth.

The explicit expression used is:

$$\frac{dB}{dt} = \mu_{\max} B \left( \frac{S}{K_S + S} \right) \left( \frac{\tanh k(T - x_0)}{1 + c(T - x_{\text{decay}})^4} \right) \exp\left( -|\text{pH} - \text{pH}_{\text{opt}}| \right) \sin\left( \frac{(\text{pH} - \text{pH}_{\min})\pi}{\text{pH}_{\max} - \text{pH}_{\min}} \right)^2$$

This formulation introduces complex nonlinearities and multimodal interactions across environmental axes, challenging models to integrate biological structure while avoiding rote memorization.

## D.3  MATERIAL STRESS BEHAVIOR

To evaluate symbolic models under realistic experimental conditions, this benchmark focuses on the stress-strain response of Aluminum 6061-T651 under thermal influence. The dataset records tensile strength measurements at six different temperatures, ranging from room temperature to 300°C, simulating diverse material states.

In contrast to synthetic equations, this task lacks an explicit ground-truth formula, requiring data-driven inference of hidden physical laws. It serves as a test of a model's capacity to identify empirical regularities in noisy, high-variance regimes.

A widely used approximation of this behavior is given by:

$$\sigma = (A + B\varepsilon^n) \left( 1 - \left( \frac{T - T_r}{T_m - T_r} \right)^m \right)$$

where $\sigma$ denotes stress, $\varepsilon$ is strain, $T$ is the temperature, $T_r$ is a reference point, and $T_m$ is the melting point. The coefficients $A$, $B$, $n$, and $m$ are empirically determined for the alloy.

This benchmark emphasizes the need for robust symbolic reasoning in the absence of prior symbolic templates, bridging theory-driven modeling with experimental data interpretation.

## D.4 LSR-SYNTH–CHEMISTRY

The **LSR-Synth–Chemistry** dataset, introduced as part of the LLM-SRBENCH benchmark by Shojaee et al., is designed to evaluate symbolic regression models on chemically motivated yet synthetically constructed reaction kinetics. It comprises 36 differential equation discovery tasks, each modeling the time evolution of a reactant concentration $A(t)$ using novel, data-driven expressions.

Each equation features a distinct combination of symbolic components—ranging from classic kinetic motifs (e.g., first- and second-order decay terms like $-kA(t)$, $-kA(t)^2$) to synthetic nonlinearities. These include exponential decays such as $\exp(-k_s t)$, logarithmic forms like $\log(A(t)+1)$, square roots, and oscillatory terms such as $\sin(\omega A(t))$ and $\cos(\cdot)$. In addition, rational expressions like $\frac{A(t)^2}{1+\beta A(t)^4}$ introduce challenges in terms of singularity avoidance and numerical stability.

The dataset emphasizes symbolic diversity and parametric variability, making it a rigorous testbed for evaluating model generalization, interpretability, and robustness across structurally distinct formulations. Each task was carefully validated for analytical solvability, numerical stability, and scientific plausibility via expert review. The full list of governing equations is available in Table 3.

Table 3: LSR-Synth Chemistry Equations (CRK1-CRK36)

| Equation ID | Equation |
|---|---|
| CRK1 | $-kA(t)^2 + k_z A(t)^2/(\beta A(t)^4 + 1)$ |
| CRK2 | $-kA(t)^2 - kA(t) + k_w \cos(\log(A(t)+1))$ |
| CRK3 | $-kA(t) + k_w \cos(\log(A(t)+1))$ |
| CRK4 | $-kA(t)^2 - kA(t)\exp(-k_s t) + k_w \cos(\log(A(t)+1))$ |
| CRK5 | $-kA(t)^2 + k_q A(t)\log(\gamma t + 1)$ |
| CRK6 | $-k\sqrt{A(t)} + k_f A(t)^{0.33}$ |
| CRK7 | $-kA(t)\exp(-k_s t) + k_m \sin(\sqrt{A(t)})$ |
| CRK8 | $-kA(t)\exp(-k_s t) + k_w \cos(\log(A(t)+1))$ |
| CRK9 | $-kA(t)^2 - kA(t) + k_t \sin(\log(A(t)+1))$ |
| CRK10 | $-k\sqrt{A(t)} + k_w \cos(\log(A(t)+1))$ |
| CRK11 | $-kA(t)^2 + k_t \sin(\log(A(t)+1))$ |
| CRK12 | $-kA(t)^2 + k_m \sin(\sqrt{A(t)})$ |
| CRK13 | $-kA(t)\exp(-k_s t) + k_t \sin(\log(A(t)+1))$ |
| CRK14 | $-kA(t) + k_p \sin(\omega A(t))$ |
| CRK15 | $-k\sqrt{A(t)} - kA(t)\exp(-k_s t) + k_p \sin(\omega A(t))$ |
| CRK16 | $-kA(t)^2 - kA(t)\exp(-k_s t) + k_t \sin(\log(A(t)+1))$ |
| CRK17 | $-kA(t) + k_f A(t)^{0.33}$ |
| CRK18 | $-kA(t)\exp(-k_s t) + k_f A(t)^{0.33}$ |
| CRK19 | $-kA(t)^2 + k_p \sin(\omega A(t))$ |
| CRK20 | $-kA(t)^2 - kA(t)\exp(-k_s t) + k_t \sin(\log(A(t)+1))$ |
| CRK21 | $-kA(t)\exp(-k_s t) + k_p \sin(\omega A(t))$ |
| CRK22 | $-kA(t)^2 + k_q A(t)\log(\gamma t + 1)$ |
| CRK23 | $-kA(t)^2 - kA(t)\exp(-k_s t) + k_z A(t)^2/(\beta A(t)^4 + 1)$ |
| CRK24 | $-k\sqrt{A(t)} + k_p \sin(\omega A(t))$ |
| CRK25 | $-k\sqrt{A(t)} - kA(t)^2 + k_f A(t)^{0.33}$ |
| CRK26 | $-kA(t) + k_t \sin(\log(A(t)+1))$ |

Table 3 – continued from previous page

| Equation ID | Equation |
|---|---|
| CRK27 | $-kA(t)^2 - kA(t)\exp(-k_s t) + k_m \sin(\sqrt{A(t)})$ |
| CRK28 | $-kA(t)^2 - kA(t)\exp(-k_s t) + k_f A(t)^{0.33}$ |
| CRK29 | $-kA(t)\exp(-k_s t) + k_z A(t)^2/(\beta A(t)^4 + 1)$ |
| CRK30 | $-kA(t) - kA(t)\exp(-k_s t) + k_z A(t)^2/(\beta A(t)^4 + 1)$ |
| CRK31 | $-kA(t) - kA(t)\exp(-k_s t) + k_t \sin(\log(A(t) + 1))$ |
| CRK32 | $-k\sqrt{A(t)} - kA(t) + k_w \cos(\log(A(t) + 1))$ |
| CRK33 | $-kA(t) - kA(t)\exp(-k_s t) + k_f A(t)^{0.33}$ |
| CRK34 | $-k\sqrt{A(t)} - kA(t)^2 + k_t \sin(\log(A(t) + 1))$ |
| CRK35 | $-kA(t)^2 + k_f A(t)^{0.33}$ |
| CRK36 | $-kA(t) + k_q A(t)\log(\gamma t + 1)$ |

# E  ADDITIONAL ABLATION STUDIES

## E.1  ABLATION STUDY ON REMAINING BENCHMARKS

We further validated the key components via ablations on the remaining three benchmarks, with the results shown in Figure 7 to Figure 9. The experimental results are highly consistent with the findings presented in the main text. First, for the single-objective optimization variant (w/o Multi-Obj), a significant degradation in out-of-domain (OOD) generalization ability was observed across all benchmarks when compared to the full model. Although the in-domain (ID) fitting error of this variant is comparable to that of the full TAMOSR on some tasks, its performance on OOD data systematically confirms the tendency of single-objective optimization to overfit the ID data. This tendency consequently hinders the discovery of universally applicable scientific laws.

Second, the ablation experiments on the internal components of the Meta Strategy Generator show that removing either the data analysis module (w/o Data) or the structure analysis module (w/o Struct) leads to a distinct decline in model performance. More critically, the degree of performance degradation caused by removing both modules simultaneously (w/o Strategy) exceeds the impact of removing either single module. This phenomenon clearly reveals the functional complementarity and synergistic effect between the data-driven prior knowledge and the structure-driven feedback mechanism. Both are indispensable for achieving TAMOSR's final performance.

## E.2  ABLATION STUDY ON DATA ANALYSIS TOOLS

To evaluate the contributions of the data analysis tools in TAMOSR, we conducted a series of ablation experiments. Our framework integrates six categories of analysis tools, and we set up six control groups (w/o tool-1 to 6), each corresponding to the removal of one category of tools.

As shown in the Figure 10, removing any single category of analysis tools in this test leads to a decline in the model's final performance. It is noteworthy that although there is only one tool in the sixth category, its removal still resulted in a final error nearly an order of magnitude higher than that of the full TAMOSR. This result strongly demonstrates that external analysis tools can effectively enhance a large language model's insight into the underlying structure and variable relationships within data, enabling it to capture deeper trends behind the data points and thereby generate more accurate symbolic equations.

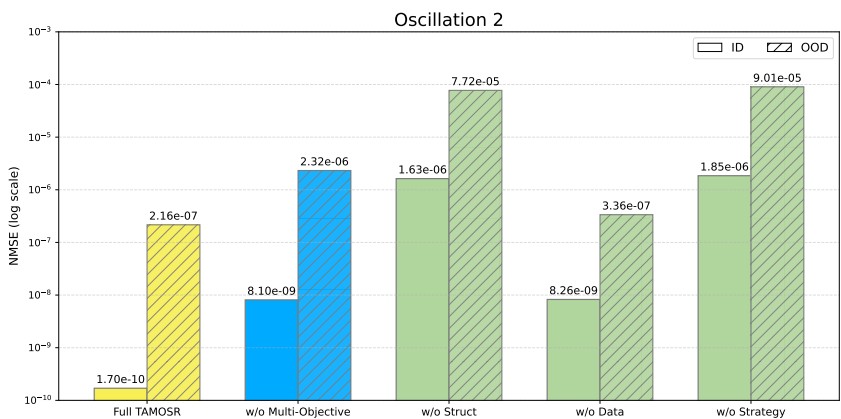

Figure 7: Ablation study on Oscillation 2.

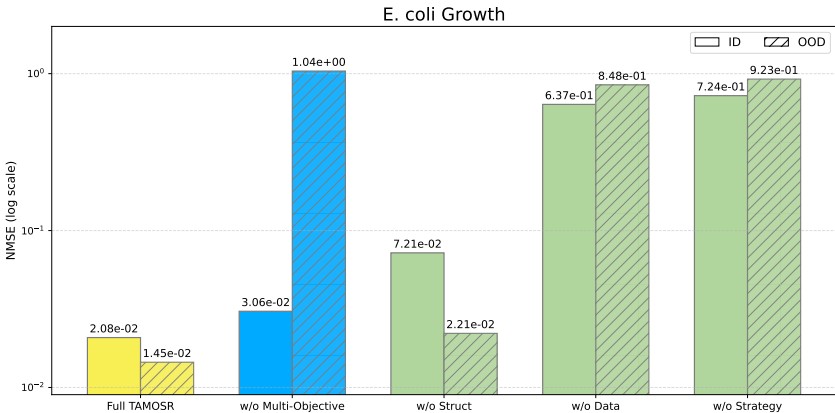

Figure 8: Ablation study on E. coli growth.

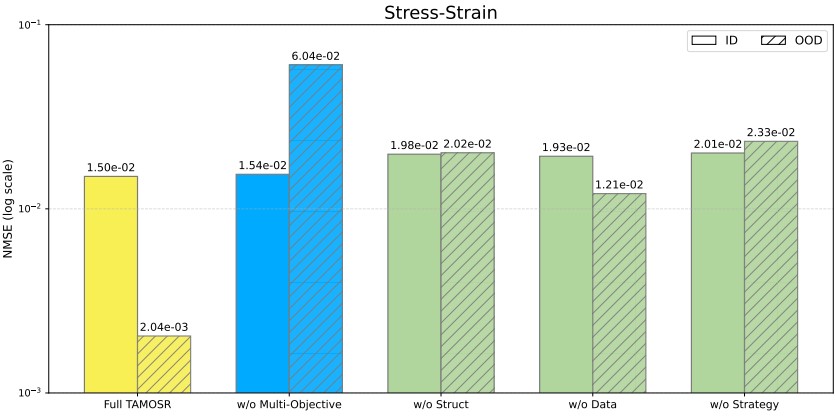

Figure 9: Ablation study on Stress-strain.

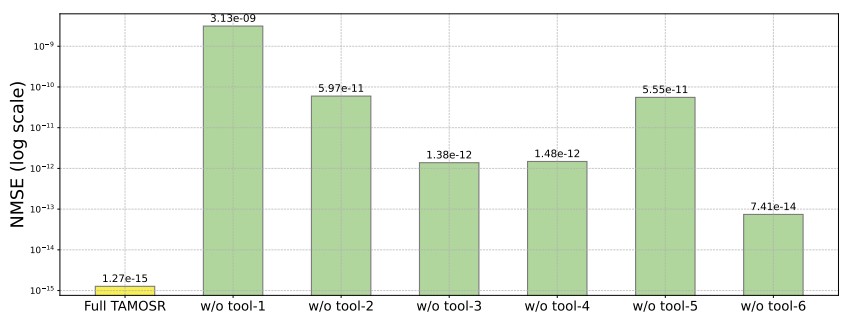

Figure 10: Ablation study of the Data Analysis Tools on Oscillator 1.

# F  HV AND IGD DEFINITION

**F.1 HV**   The hypervolume (HV) measures the volume in the objective space that is dominated by the approximate Pareto front and bounded by a reference point. It reflects the convergence and diversity of the solutions. The HV is formally defined as follows:

$$\mathrm{HV}(P, r^*) = \mathrm{VOL}\left(\bigcup_{\mathbf{v} \in P} [v_1, r_1^*] \times [v_2, r_2^*] \times \cdots \times [v_m, r_m^*]\right),$$

where $P$ denotes the approximate Pareto front obtained by the symbolic regression algorithm, $\mathbf{v} = (v_1, \ldots, v_m)^\top$ represents an objective vector, $\mathrm{VOL}(\cdot)$ indicates the Lebesgue measure, and $r^* = (r_1^*, \ldots, r_m^*)^\top$ is the reference point.

To eliminate the impact of varying scales and units among objectives in HV calculation, we normalize each objective to the interval $[0, 1]$. For an objective value $f_i(x)$, the normalized value is:

$$f_i'(x) = \frac{f_i(x) - z_i^{\mathrm{ideal}}}{z_i^{\mathrm{nadir}} - z_i^{\mathrm{ideal}}},$$

where

$$z_i^{\mathrm{ideal}} = \min_{\mathbf{v} \in P} v_i, \quad z_i^{\mathrm{nadir}} = \max_{\mathbf{v} \in P} v_i$$

After normalization, each objective is scaled to the range $[0, 1]$. Based on this, the reference point is set as $r^* = (1.05, \ldots, 1.05)^\top$.

**F.2 IGD**   The Inverted Generational Distance (IGD) measures both the convergence and diversity of the predicted front by computing the distance from points on the true Pareto front to their nearest counterparts in the predicted set. The formula is as follows:

$$\mathrm{IGD}(P, P^*) = \frac{1}{|P^*|} \sum_{\mathbf{p}^* \in P^*} \min_{\mathbf{p} \in P} d(\mathbf{p}, \mathbf{p}^*),$$

where $P$ is the predicted Pareto front, $P^*$ is a set of reference points sampled from the true Pareto front, and $d(\mathbf{p}, \mathbf{p}^*)$ represents the Euclidean distance between vectors $\mathbf{p}$ and $\mathbf{p}^*$ in objective space.

The computation of the Inverted Generational Distance (IGD) metric conventionally depends on the availability of a known true Pareto front, which is not always accessible in practical scenarios. In the context of symbolic regression evaluation, however, the IGD metric benefits from an inherent advantage, as the true Pareto front can be explicitly defined by the ground truth equation $f^{\mathrm{true}}$. This allows for an exact assessment of the quality of generated equations relative to the ground truth,

without reliance on an approximate Pareto front. Accordingly, the IGD formulation in this study is simplified as follows:

$$\text{IGD}(P, \{f^{\text{true}}\}) = \min_{f \in P} d(f, f^{\text{true}})$$

Afterwards, all objectives are normalized to the $[0, 1]$ range to ensure consistency with the true objective values.

## G   DATA ANALYSIS TOOLS IN TAMOSR

### G.1   LINEAR CORRELATION TOOLS

Linear correlation tools provide foundational insights into how variables are related through linear or approximately linear patterns. These tools are instrumental for identifying primary dependencies, detecting dominant axes of variation, and establishing structural priors for symbolic equation modeling. TAMOSR incorporates four linear-correlation-based modules:

- **Pearson Correlation Coefficient:**
  It quantifies the degree of linear association between two variables $x$ and $y$ using the classical Pearson correlation coefficient $r$. The input consists of two real-valued vectors, the output includes the correlation score and the associated $p$-value indicating statistical significance. A high absolute value of $r$ (close to 1) suggests a strong linear trend, while values near 0 indicate little to no linear dependency. This measure is especially suitable for Gaussian-like distributions and helps identify candidate variables for symbolic terms with additive or multiplicative linear effects.

- **Simple Linear Regression:**
  This module fits a univariate linear model $y = a + bx$ using least squares estimation and returns the regression coefficients, the $R^2$ value (explained variance), and associated statistics such as $p$-values and standard errors. The inputs are one-dimensional arrays $x$ and $y$, and the output reflects how well a straight line fits the data. The $R^2$ value is particularly informative, indicating how much of the variance in $y$ can be explained by $x$. It provides a predictive perspective on the relationship beyond correlation, supporting model selection based on explanatory power.

- **Residual Variance:**
  This tool computes the variance of residuals $y - \hat{y}$ after applying linear regression. The lower the residual variance, the better the linear model captures the relationship between $x$ and $y$. It takes the same inputs as Simple Linear Regression and internally reuses the linear regression output. This metric emphasizes prediction error dispersion, offering a direct measure of the noise or unmodeled nonlinear structure in the data. It is particularly valuable for pruning noisy or unstable variables from candidate model terms.

- **PCA Explained Variance:**
  Principal Component Analysis (PCA) is applied to the joint space of $x$ and $y$, and the proportion of variance explained by the first principal component is reported. The input is a two-dimensional matrix formed by stacking $x$ and $y$, the output is a scalar value indicating the strength of the shared linear structure. This approach is robust to scaling and rotation and can capture the dominant direction of variation when the relationship is more general than univariate regression. It aids in identifying latent linear couplings and guides the selection of structurally informative variables.

### G.2   NONLINEAR DEPENDENCY TOOLS

Nonlinear dependency tools are employed to capture complex, non-monotonic interactions between variables, which are essential for accurately modeling nonlinear systems. Unlike linear tools, these methods can reveal relationships that are not adequately described by simple linear transformations. TAMOSR incorporates three nonlinear-dependency-based modules:

- **Spearman Rank Correlation:** This tool measures the strength and direction of monotonic relationships between two variables using their rank values. It computes Spearman's $\rho$, a non-parametric counterpart to Pearson's $r$, by evaluating how well the relationship between $x$ and $y$ can be described by a monotonic function. The input consists of two numerical vectors, the output

includes the correlation coefficient and its $p$-value. This method is particularly effective when data exhibit nonlinear but monotonic trends, such as saturating growth or sigmoid-like behavior.

- **Mutual Information:** This tool quantifies the total amount of information shared between two variables, regardless of the specific functional form of their relationship. It estimates mutual information by discretizing the input variables $x$ and $y$ into bins and calculating the joint entropy. The output is a scalar metric representing dependency strength. Unlike correlation coefficients, mutual information can capture both linear and highly nonlinear dependencies, making it suitable for detecting complex statistical associations in symbolic modeling.

- **Mutual Information Regression Score:** This variant employs the `scikit-learn` implementation of mutual information to assess the relevance of $x$ in predicting $y$ using a regression-based formulation. The method internally estimates how much knowing $x$ reduces uncertainty about $y$. It accepts one-dimensional arrays as input and outputs a scalar score. This score is robust to arbitrary nonlinearities and discontinuities, making it valuable for feature selection in nonlinear symbolic regression tasks.

### G.3 TIME-FREQUENCY ANALYSIS TOOLS

Time-frequency analysis tools are essential for detecting periodicities, oscillations, and transient dynamics that are characteristic of nonlinear and multi-scale systems. These tools enable symbolic regression models to incorporate periodic or time-varying terms where appropriate. TAMOSR employs two such modules:

- **Fast Fourier Transform Frequency Difference:** This module applies the discrete Fourier transform to the input signals $x$ and $y$ to extract their respective power spectra. It identifies the dominant frequency component for each variable and returns the absolute difference between them. The input consists of two one-dimensional arrays representing time-series data, and the output includes the dominant frequencies of both variables as well as their difference. This tool captures global periodic patterns and is useful for detecting whether the signals share synchronized or harmonically related structures. A small frequency difference suggests potential functional alignment via sinusoidal or oscillatory terms.

- **Wavelet Energy Correlation:** This module leverages discrete wavelet decomposition to capture localized energy features of $x$ and $y$ at multiple temporal scales. It decomposes both signals into several levels using a specified wavelet basis (e.g., Daubechies 4), computes the energy at each level, and measures the Pearson correlation between the resulting energy vectors. The input includes two one-dimensional arrays and optional parameters for wavelet type and decomposition level. The output is a correlation coefficient representing how similarly the energy of the two signals is distributed across scales. Unlike Fourier-based methods, this tool excels at capturing transient and non-stationary dependencies, offering robust structural priors for equations involving local periodicity or bursts.

### G.4 CAUSAL INFERENCE TOOLS

Causal inference tools aim to uncover directional relationships between variables, particularly whether the historical behavior of one variable contributes to or influences another. These tools help TAMOSR to build models that not only fit the data well but also respect temporal or structural causality, thereby improving interpretability and generalization. Two modules are implemented:

- **Granger Causality:** This method assesses whether past values of a variable $x$ improve the prediction of a variable $y$ in a multivariate time series setting. The inputs are two equal-length time series arrays, and the test evaluates multiple lagged regression models to determine if $x$ Granger-causes $y$. The output includes $p$-values for different lag orders, and the smallest $p$-value is used as the primary metric. A statistically significant result implies that the past of $x$ contains information predictive of $y$, suggesting a directional dependency. This method is especially valuable for systems where delayed effects are prominent, such as in control dynamics or feedback loops.

- **Convergent Cross Mapping:** CCM is a nonlinear causal discovery technique grounded in dynamical systems theory. It tests whether the state of variable $x$ can be reconstructed from the historical trajectory of $y$, indicating that $x$ leaves an imprint on $y$. The input consists of two time

series and parameters for embedding dimension and library sampling. CCM constructs a manifold from the delay embedding of $y$, uses it to cross-predict $x$, and evaluates the reconstruction accuracy (typically using a cross-map skill $\rho$). A high $\rho$ suggests that $x$ causally influences $y$. Unlike Granger causality, CCM does not rely on linearity or temporal precedence, making it ideal for identifying nonlinear, feedback-driven causality in complex systems.

### G.5 DYNAMIC COMPLEXITY TOOLS

Dynamic complexity tools capture nuanced structural and temporal irregularities in time-series data, particularly useful for distinguishing chaotic, nonlinear, or asynchronous behaviors. TAMOSR leverages three such tools:

- **Lyapunov Exponent Difference:** This tool estimates the maximal Lyapunov exponents (MLEs) of two time-series variables $x$ and $y$, then computes their absolute difference. Lyapunov exponents characterize how sensitive a system is to initial conditions. A high MLE suggests chaotic behavior, whereas near-zero or negative values indicate stable dynamics. The input consists of two real-valued time series. The tool automatically determines the embedding dimension and time lag for phase space reconstruction. The resulting metric reflects how similarly (or differently) $x$ and $y$ behave in terms of dynamical predictability.

- **Correlation Dimension Difference:** This tool quantifies and compares the fractal (correlation) dimensions of $x$ and $y$. The correlation dimension serves as a complexity measure, indicating how densely a system's trajectory fills its phase space. The input includes two real-valued sequences and a fixed embedding dimension (usually 2). The metric is the absolute difference between their estimated correlation dimensions. This captures differences in dynamic complexity and is useful for identifying structural mismatches in multivariate nonlinear processes.

- **Dynamic Time Warping Distance:** DTW measures the alignment cost between $x$ and $y$ by allowing local nonlinear stretching or compression in time. Unlike Euclidean distance, DTW is robust to phase shifts and unequal pacing in temporal evolution. The input is a pair of univariate time series, and the output is a scalar DTW distance. This tool is particularly effective in identifying temporal patterns that share similar shapes but occur at different rates or phases.

### G.6 DISTRIBUTION CONSISTENCY TOOLS

This category includes statistical tools that assess whether two variables share similar probability distributions. Such tools are especially useful when validating whether a generated or transformed signal preserves the underlying distributional structure of the original data.

- **Kolmogorov–Smirnov Test:** The Kolmogorov–Smirnov (KS) test is a non-parametric method that quantifies the maximum distance between the empirical cumulative distribution functions (ECDFs) of two datasets $x$ and $y$. The input to this tool is a pair of real-valued vectors, and it returns the KS statistic (a scalar metric) along with the corresponding $p$-value indicating statistical significance. The KS statistic captures how different the two distributions are—larger values indicate more significant deviations. This test is sensitive to both location and shape differences in the distributions and is thus useful for evaluating whether symbolic transformations (e.g., derived equations) maintain statistical fidelity to the data source.

## H RESULTS PRESENTATION

### H.1 EQUATION RECOVERY AND PARETO FRONT EVOLUTION

Fig 11a to Fig 11d are sampled from four test cases in which TAMOSR successfully identified the ground-truth equations on the LSR-Synth–Chemistry dataset. Each figure illustrates the evolution of NMSE of the Pareto front equation set across iterations. For visualization, we present the structural skeletons of the equations discovered by TAMOSR. Highlighted segments in red indicate terms that match exactly with the corresponding ground-truth equation.

It is noteworthy that the size of the Pareto front typically expands rapidly during the early stages, forming a diverse equation population. As iterations proceed, this population undergoes continuous

updates. Once sufficient structural experience has accumulated, the Pareto front enters a phase of rapid convergence and eventually stabilizes to the correct target equation. Below we present comparisons between TAMOSR and LLM-SR on the same tasks:

**CRK14:**

Ground truth:
$$-kA(t) + k_p \sin(\omega A(t))$$

TAMOSR:
$$\boxed{\theta_0 A} + \boxed{\theta_1 \sin(\theta_2 A)}$$

LLM-SR:
$$\frac{\theta_0 A \left(1 + \frac{(\theta_1 A)^2}{\theta_2}\right)}{1 + \frac{1 + \frac{A^2}{\theta_5}}{1 + \frac{(\theta_1 A)^2}{\theta_2}}} + \frac{\theta_4 A}{\theta_5 + A} + \boxed{\theta_3 A}$$

**CRK19:**

Ground truth:
$$-kA(t)^2 + k_p \sin(\omega A(t))$$

TAMOSR:
$$\boxed{\theta_0 \sin(\theta_1 A)} + \boxed{\theta_2 A^2}$$

LLM-SR:
$$-\theta_0 A - \boxed{\theta_1 A^2} + \theta_2 t + \theta_3 t^2 + \theta_4 A^3$$

**CRK21:**

Ground truth:
$$-kA(t)e^{-k_s t} + k_p \sin(\omega A(t))$$

TAMOSR:
$$\boxed{\theta_0 A e^{-\theta_1 t}} + \boxed{\theta_2 \sin(\theta_3 A)}$$

LLM-SR:
$$\theta_0 t + \boxed{\theta_1 A e^{-\theta_2 t}} + \frac{\theta_3 A}{\theta_4 t + 1} + \frac{\theta_5 A^{\theta_6}}{1 + \theta_7 A^{\theta_6}}$$

**CRK36:**

Ground truth:
$$-kA(t) + k_q A(t) \log(\gamma t + 1)$$

TAMOSR:
$$\boxed{\theta_0 A} + \boxed{\theta_1 A \log(\theta_2 t + 1)}$$

LLM-SR:
$$\boxed{\theta_0 A} + \theta_1 e^{-\theta_2 t} A + \frac{\theta_3 A}{\theta_4 + A} + \frac{\theta_5 A}{1 + \theta_6 A} + \theta_7 t + \boxed{\theta_8 A} + \theta_9 t A$$

These results clearly demonstrate that TAMOSR is capable of precisely capturing the exact components of the ground-truth equations, yielding structurally closer expressions than LLM-SR. We attribute this improvement to TAMOSR's enhanced exploratory capability derived from multi-objective evaluation, as well as its strategy-guided generation process.

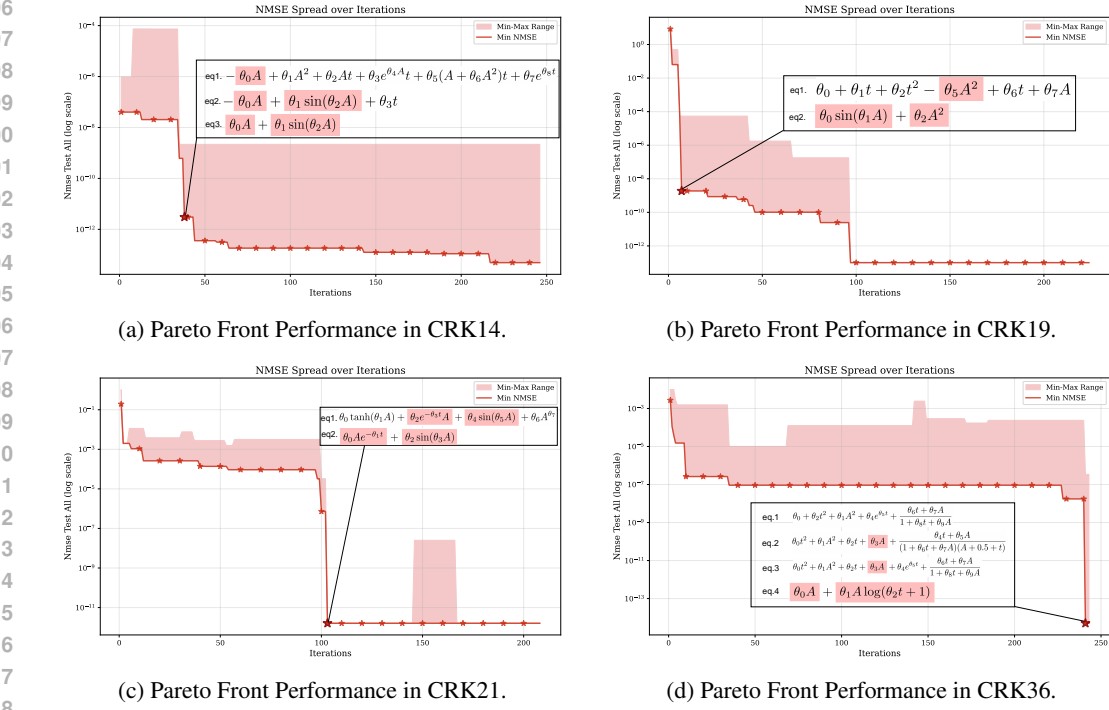

(a) Pareto Front Performance in CRK14.

(b) Pareto Front Performance in CRK19.

(c) Pareto Front Performance in CRK21.

(d) Pareto Front Performance in CRK36.

Figure 11: Pareto Front Performance results across four benchmarks.

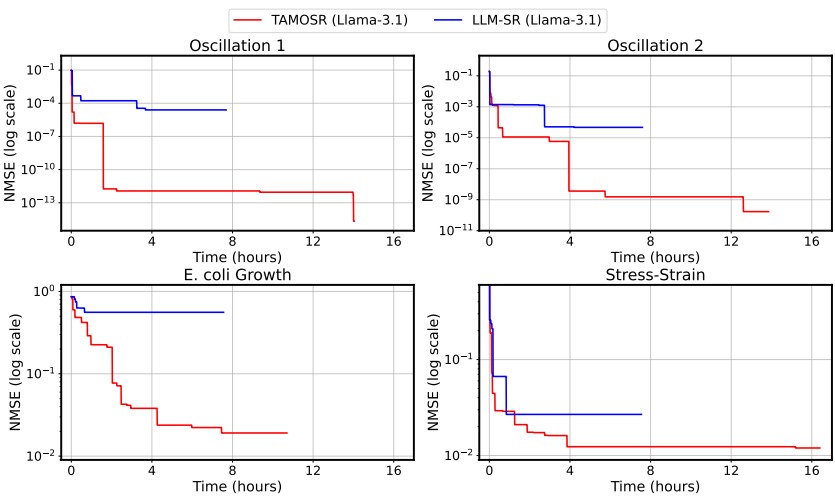

Figure 12: Wall-clock time evolution of NMSE showing faster convergence of TAMOSR over LLM-SR.

### H.2 COMPARISON OF CONVERGENCE DYNAMICS.

Figure 12 presents the wall-clock time evolution of NMSE (log scale) for TAMOSR and the baseline LLM-SR across four representative benchmarks. The results demonstrate that TAMOSR consistently achieves substantially faster error reduction, while LLM-SR converges more slowly and plateaus at significantly higher error levels. This advantage is particularly pronounced in the Oscillation tasks, where TAMOSR rapidly identifies accurate dynamical structures, but is also evident in more challenging domains such as *E. coli* growth and stress–strain modeling, where the baseline stagnates prematurely. These findings highlight that the integration of tool-guided analysis and

multi-objective optimization enables TAMOSR to explore the hypothesis space more efficiently, yielding superior equations under comparable computational budgets.

## H.3 EVOLUTION OF DISCOVERED EQUATIONS

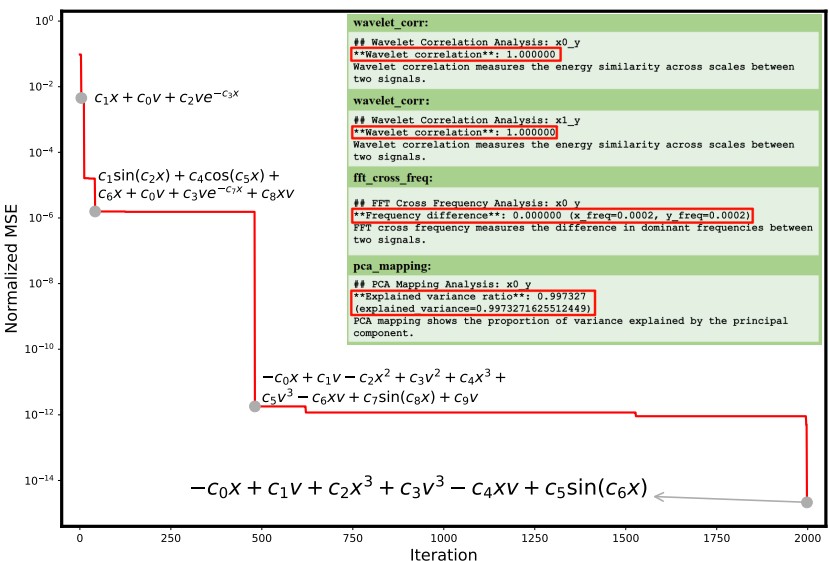

Figure 13: Iterative evolution of equation structures under TAMOSR (Oscillation1).

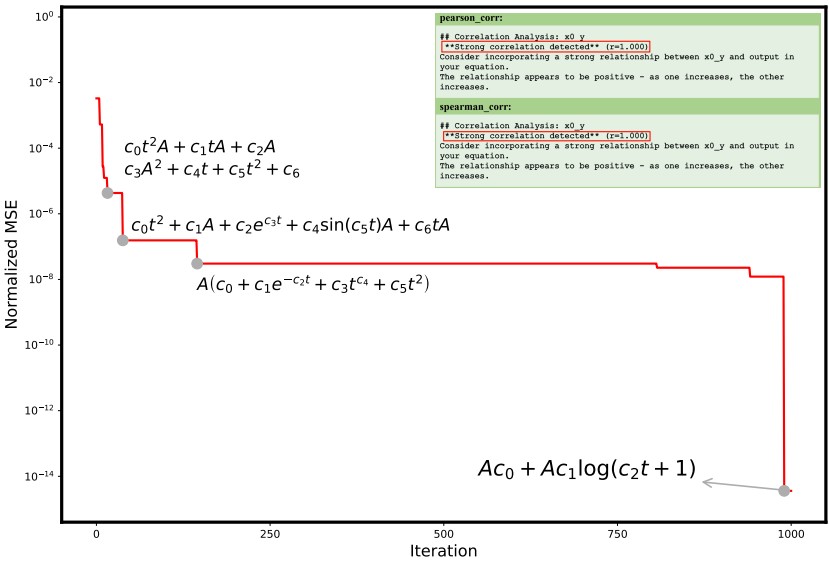

Figure 14: Iterative evolution of equation structures under TAMOSR (CRK36).

**Evolution of Equation Structures for Oscillation 1**    Figure 13 presents the iterative evolution of equation structures for the problem oscillation 1. In the early iterations, the generated expressions contained a wide range of nonlinear and polynomial terms, many of which represented exploratory hypotheses rather than meaningful components. Guided by the TOOL-AUGMENTED VARIABLE ANALYSIS module, the framework progressively refined these structures. When the analytical tools within this module (e.g., wavelet corr, FFT, and PCA mapping) revealed periodic features,

sinusoidal terms such as sine and cosine functions were introduced into the candidate equations, thereby complementing the polynomial basis and improving alignment with the observed data.

During the iteration, structurally inconsistent or statistically unsupported terms were systematically removed, while the retained nonlinear and oscillatory components captured the essential dynamics of the system. The process ultimately converged toward compact formulations that balance accuracy, generalization, and complexity. This refinement demonstrates how the iterative mechanism, augmented by TOOL-AUGMENTED VARIABLE ANALYSIS, effectively distills the equation space and produces models consistent with both empirical behavior and theoretical plausibility.

**Evolution of Equation Structures for CRK36**   Figure 14 presents the iterative evolution of equation structures for the problem CRK36. At the beginning of the process, the candidate functions contained a variety of nonlinear combinations involving both the temporal variable and A. Under the guidance of two correlation coefficients, the framework progressively refined these structures by eliminating inconsistent nonlinearities and retaining only the dominant dependencies. As the iterations advanced, the generated functions converged toward a fully linear dependence on A.

# I   GAUSSIAN NOISE EVALUATION

To evaluate the robustness of TAMOSR in noisy environments, we conduct experiments on the `Oscillator1` benchmark by injecting Gaussian noise with standard deviations $\sigma = \{0.001, 0.002\}$ into the training data. Model performance is assessed under both in-domain (ID) and out-of-domain (OOD) test conditions.

Unlike conventional symbolic regression approaches that rely solely on fitting accuracy, TAMOSR integrates structural priors—derived from analytical tools—and performs multi-objective evaluation that jointly considers accuracy, complexity, and generalization. This allows it to maintain high-fidelity symbolic recovery even when data are corrupted.

Table 4: Comparison of TAMOSR and LLMSR on `Oscillator1` under Gaussian noise ($\sigma = \{0.001, 0.002\}$). Metrics include NMSE and accuracy ($\text{Acc}_{\text{avg-0.01}}$) on ID and OOD test sets. All models use LLaMA-3.1 as backbone.

| Model | $\sigma = 0.001$ | | | | $\sigma = 0.002$ | | | |
|---|---|---|---|---|---|---|---|---|
| | ID | | OOD | | ID | | OOD | |
| | NMSE↓ | $\text{Acc}_{\text{avg-0.01}}$↑ | NMSE↓ | $\text{Acc}_{\text{avg-0.01}}$↑ | NMSE↓ | $\text{Acc}_{\text{avg-0.01}}$↑ | NMSE↓ | $\text{Acc}_{\text{avg-0.01}}$↑ |
| **LLMSR** | $1.59e\text{-}5$ | $89.20\%$ | $0.0021$ | $10.46\%$ | $2.68e\text{-}5$ | $82.08\%$ | $0.0025$ | $10.50\%$ |
| **TAMOSR** | **5.43e-11** | **99.99%** | **5.49e-7** | **98.79%** | **3.98e-8** | **99.74%** | **8.57e-7** | **97.88%** |

Table 4 shows that TAMOSR consistently outperforms LLMSR across all metrics, achieving significantly lower NMSE and higher $\text{Acc}_{\text{avg-0.01}}$ scores under both noise settings. These gains stem from TAMOSR's integration of tool-informed structural priors via the Meta Strategy Generator and its multi-objective Equation Generator, which jointly enable the model to identify robust, generalizable expressions even under noisy supervision.

# J   PROMPT

**J.1 Prompt Design**   The prompts used to construct the meta strategy are shown in Fig. 15 and Fig. 16. Specifically, Fig. 15 presents the meta prompt used during initialization, while Fig. 16 corresponds to the prompts designed for iterative optimization. Together, these constitute the prompt inputs to the Meta Strategy Generator.

**J.2 Examples**   The prompts used by the Equation Generator for the Oscillators 1 task are illustrated in Fig. 17 to Fig. 20. The prompts received by the Meta Strategy Generator for the same task are shown in Fig. 21 to Fig. 26.

Fig. 27 provides examples of data analysis tool usage, drawn from tool invocations in the CRK36 task. Notably, through the integration of external tools, the language model successfully identified a strong linear relationship between the variable $x_0$ (that is, $A(t)$) and the target variable $y$. As demonstrated in the final output, TAMOSR successfully recovered the ground-truth governing equation for the CRK36 problem.

---

**Meta Prompt**

```
SYSTEM ROLE: You are a **scientist-assistant LLM** supervising a
symbolic-regression pipeline.

You must analyze existing equations, select appropriate analysis tools,
and produce guidance for a generator LLM.

## Allowed analysis tools
{allowed_list}
Choose **up to 3** tools—exclusively from the list above—based **solely
on the diagnostic needs revealed by the residual data you receive**.If a
desired tool is not listed, omit it.

## RESPONSE FORMAT (STRICT)
Return exactly one JSON object (no markdown fences). It must contain:
1. "structure_insight_prompt_for_generator_LLM" – either a string, or
   an object with `guidance.prompt_for_generator_LLM` (string).
2. "analysis_tools" – a JSON array of 1-5 tool names from the allowed
list.

Escape newlines inside strings as `\n`. After the closing brace `}`,
output nothing else.
```

Figure 15: Meta Prompt.

---

**Refinement Prompt**

```
## Context
{context}

## Mathematical Pattern Analysis Task
Analyze the following equations to identify **common mathematical
patterns and shared structural elements**:

.join(blocks)
## Available Analysis Tools
{tool_list}
## Pattern Analysis Instructions
1. **Examine Mathematical Structures**: Look across all equations for:
   - Recurring mathematical functions (trigonometric, exponential,
polynomial)
   - Similar variable interaction patterns (x*y, x+y, x^n, etc.)
   - Consistent coefficient magnitudes or sign patterns
   - Common denominators or fraction structures
   - Similar nesting or grouping of terms
```

```
.join(blocks)
## Available Analysis Tools
{tool_list}
## Pattern Analysis Instructions
1. **Examine Mathematical Structures**: Look across all equations for:
   - Recurring mathematical functions (trigonometric, exponential,
polynomial)
   - Similar variable interaction patterns (x*y, x+y, x^n, etc.)
   - Consistent coefficient magnitudes or sign patterns
   - Common denominators or fraction structures
   - Similar nesting or grouping of terms

2. **Identify Shared Elements**: Find mathematical components that appear
in multiple high-performing equations

3. **Interpret Mathematical Significance**: Explain what these patterns
might indicate about:
   - Underlying physical relationships
   - Mathematical constraints or principles
   - Optimal solution characteristics

4. **Generate Guidance**: Create a detailed prompt that will guide future
equation generation based on successful patterns

5. **Select Analysis Tools**: Choose up to 5 data analysis tools that
could help validate the patterns you identified

## RESPONSE FORMAT
Return ONLY JSON with keys: structure_insight_prompt_for_generator_LLM,
analysis_tools.
- structure_insight_prompt_for_generator_LLM: Detailed guidance based on
your pattern analysis for generating better equations
- analysis_tools: Array of up to 5 tool names from the available list

Focus on mathematical commonalities and their significance in equation
generation.
```

Figure 16: Prompts Designed for Iterative Optimizationt.

```
Oscillator1

You are a helpful assistant tasked with discovering mathematical function
structures for scientific systems.
Complete the 'equation' function below, considering the physical meaning
and relationships of inputs.

# Scientific Analysis Based on Pareto Front Selection

## Core Generation Guidance
## Core Generation Guidance
Generate equations that incorporate the identified mathematical patterns
and shared structural elements. Emphasize leveraging variable interactions,
consistent coefficient magnitudes, and common denominators. Prioritize
equations that respect physical constraints and optimization
characteristics.

## Generation Instructions
Based on the above analysis, focus on:
1. Incorporating the identified recurring mathematical functions
```

```
2. Using the successful variable interaction patterns
3. Maintaining coefficient relationships that work well
4. Respecting the physical constraints and relationships
5. Building upon the shared mathematical elements
```

Figure 17: Example Prompt in Oscillator1 - For Equation Generator.

```
## Data Analysis Results
Applied 3 analysis tools to all variable pairs:

**Tools used**: spearman_corr, residual_var, corr

**Dataset overview**:
- 2 independent variables analyzed
- 10000 total data points

## Correlation Analysis: x0_y
 **Strong correlation detected** (r=-0.920)
Consider incorporating a strong relationship between x0_y and output in
your equation.
The relationship appears to be negative - as one increases, the other
decreases.

## Residual Variance Analysis: x0_y
**Residual variance**: 0.000419
Low residual variance suggests the linear relationship captures most
variation.

## Correlation Analysis: x0_y
 **Strong correlation detected** (r=-0.951)
Consider incorporating a strong relationship between x0_y and output in
your equation.
The relationship appears to be negative - as one increases, the other
decreases.

## Correlation Analysis: x1_y
 **Weak correlation detected** (r=0.103)
The relationship between x1_y and output appears weak or non-linear.

## Residual Variance Analysis: x1_y
**Residual variance**: 0.004359
Low residual variance suggests the linear relationship captures most
variation.

## Correlation Analysis: x1_y
 **Weak correlation detected** (r=-0.032)
The relationship between x1_y and output appears weak or non-linear.
```

Figure 18: Example Prompt in Oscillator1 - For Equation Generator.

```
## Generation Strategy
Based on the analysis of Pareto front samples and data relationships:

1. **Build upon success**: Use the selected high-quality solution as
foundation
2. **Targeted improvement**: Focus on reducing residuals in identified
problem areas
```

```
4. **Data-driven insights**: Incorporate findings from variable
relationship analysis
5. **Variable interactions**: Consider the significant correlations and
patterns identified
6. **Maintain balance**: Preserve the multi-objective balance that led to
Pareto selection
7. **Consider trade-offs**: Balance accuracy, generalization, and
computational efficiency

"""
Find the mathematical function skeleton that represents acceleration in a
damped nonlinear oscillator system with driving force, given data on
position, and velocity.
"""
```

Figure 19: Example Prompt in Oscillator1 - For Equation Generator.

```
def equation(x: np.ndarray, v: np.ndarray, params: np.ndarray) ->
np.ndarray:
    """ Mathematical function for acceleration in a damped nonlinear
oscillator
    Args:
        x: A numpy array representing observations of current position.
        v: A numpy array representing observations of velocity.
        params: Array of numeric constants or parameters to be optimized
    Return:
        A numpy array representing acceleration as the result of applying
the mathematical function to the inputs.
    """
    # Driving force considering correlation with x0_y and output
    f = params[0] * np.exp(-params[1] * v**2)  # Driving force with damping

    # Incorporating strong linear relationship between x0_y and output
    linear_term = params[2] * x  # Linear coefficient for damping

    # Damping effect considering correlation with x1_y and output
    damp = params[3] * (1 + 0.5 * np.sin(params[4] * x))  # Damping
coefficient with trigonometric term

    # Nonlinear oscillator equation considering correlations
    a = -(damp + 0.5 * linear_term) * v

    # Additional non-linear terms for better model fit
    b = params[5] * np.sin(params[6] * x + params[7] * v)  # Nonlinear term
with hybrid sine function
    c = params[8] * v**2 + f * np.sin(params[9] * x)  # Nonlinear term with
exponenial trig function

    # Regularization technique to prevent overfitting
    return a + b + c

def equation_v1(x: np.ndarray, v: np.ndarray, params: np.ndarray) ->
np.ndarray:
    """Improved version of `equation_v0`."""
```

Figure 20: Example Prompt in Oscillator1 - For Equation Generator.

**Oscillator1**

SYSTEM ROLE: You are a **scientist-assistant LLM** supervising a
symbolic-regression pipeline.

You must analyze existing equations, select appropriate analysis tools, and
produce guidance for a generator LLM.

## Allowed analysis tools
ccm_causality, corr, corr_dim_relation, dtw_distance, fft_cross_freq,
granger_causality, ks_test_diff, lin_reg, lyapunov_relation, mutual_info,
mutual_info_regression_score, pca_mapping, pearson_corr, residual_var,
spearman_corr, wavelet_corr

Choose **up to 3** tools—exclusively from the list above—based **solely on
the diagnostic needs revealed by the residual data you receive**.If a
desired tool is not listed, omit it.

## RESPONSE FORMAT (STRICT)
Return exactly one JSON object (no markdown fences). It must contain:
1. "structure_insight_prompt_for_generator_LLM" – either a string, or
   an object with `guidance.prompt_for_generator_LLM` (string).
2. "analysis_tools" – a JSON array of 1-5 tool names from the allowed list.

Escape newlines inside strings as `\n`. After the closing brace `}`, output
nothing else.

## Context
Symbolic regression problem: oscillator1. Dataset has 2 input variables and
1 target variable. Training data shape: (10000, 2) -> (10000,)

Analyzing equations selected from Pareto front with residual data.

## Mathematical Pattern Analysis Task
Analyze the following equations to identify **common mathematical patterns
and shared structural elements**:

Figure 21: Example Prompt in Oscillator1 - For Meta Strategy Generator.

```
### Equation 1
Expression    :     """Mathematical function for acceleration in a damped
nonlinear oscillator
    Args:
        x: A numpy array representing observations of current position.
        v: A numpy array representing observations of velocity.
        params: Array of numeric constants or parameters to be optimized
            params[0]: Spring constant, effectively the driving force
coefficient
            params[1]: Damping coefficient
            params[2]: Amplitude, the maximum value of driving force
            params[3]: Angular frequency, c=driving force frequency
            params[4]: Mass of object, m
            params[5]: Friction or driving force damping coefficient
            params[6]: Non-linear damping exponent
            params[7]: Non-linear driving force frequency millers
            params[8]: Driving force input frequency peak frequency millers
            params[9]: Non-linear displacement spring constant magnetic
field coefficients
```

```
    Returns:
        A numpy array representing acceleration as the result of applying
the mathematical function to the inputs.
    """
    # Driving force
    c = params[0]  # Original driving force coefficient
    A = params[2]  # Original amplitude

    # Frictionless surface parameters (assuming frictionless)
    m = params[4]  # mass of object

    # Non-linear ODE (On-Off Oscillators)
    # x-dependent, non-linear spring constant
    k = params[9] + params[1] * np.exp(-np.abs(v))  # Original spring
constant

    # Incorporating driving force non-linearity
    # exponential riding
    f = c * (A + np.tanh(params[6] * (v + params[7]))) * np.sin(params[8] *
x)

    # Non-linear ODE application
    return -m * (k * v + f)
```

Figure 22: Example Prompt in Oscillator1 - For Meta Strategy Generator.

```
ID Score     : -0.00013475 (in-distribution performance)
OOD Score    : -5.38816e-05 (out-of-distribution performance)
Eval Time    : 10.6924s (computational efficiency)
Avg Accuracy : -9.43156e-05 (overall accuracy)
Gen Gap      : 8.08681e-05 (generalization stability)
Efficiency   : 0.0855254 (time-normalized performance)

**In-Distribution (ID) Dataset Analysis:**
  Worst Region Statistics:
    - Sample Count: 520 samples
    - Threshold: 0.000958914
    - Mean Residual: -0.0008611
    - Mean Abs Residual: 0.00112713
    - Max Abs Residual: 0.00148016
    - Std Residual: 0.000750307
  Input Variables in Worst Region:
    x0: [-0.481813, -0.419474]    x1: [-0.124159, 0.113473]    Target
Variable (y) in Worst Region:
    y: [0.048175, 0.0951525]
**Out-of-Distribution (OOD) Dataset Analysis:**
  Worst Region Statistics:
    - Sample Count: 480 samples
    - Threshold: 0.000740516
    - Mean Residual: -0.000603334
    - Mean Abs Residual: 0.000866861
    - Max Abs Residual: 0.000997573
    - Std Residual: 0.000629749
  Input Variables in Worst Region:
    x0: [-0.520795, 0.489177]    x1: [-0.0900758, 0.282136]    Target
Variable (y) in Worst Region:
    y: [-0.0614007, 0.060305]
```

Figure 23: Example Prompt in Oscillator1 - For Meta Strategy Generator.

```
### Equation 2
Expression  :       """
    Mathematical function for acceleration in a damped nonlinear oscillator.

    Args:
        x: A numpy array representing observations of current position.
        v: A numpy array representing observations of velocity.
        params: Array of numeric constants or parameters to be optimized
            params[0]: Spring constant (k), driving force coefficient
(omega)
            params[1]: Damping coefficient (gamma)
            params[2]: Amplitude (A)
            params[3]: Angular frequency (omega)

    Returns:
        A numpy array representing acceleration as the result of applying
the mathematical function to the inputs.
    """

    # Parameters for driving force
    omega = params[0]  # Angular frequency
    omega_driving_force = omega

    # Parameters for friction or driving force damping
    gamma = params[1]  # Damping coefficient
    gamma_friction_driving_damping = gamma

    # Mass and amplitude for non-linear oscillator
    m = params[4]  # Mass
    A_driving_damping = params[2]  # Amplitude

    # Calculate velocity induced force
    force = omega_driving_force * np.sin(params[3] * np.tanh(params[5] * v))

    # Acceleration equation
    # Consider strong correlations between position and velocity
    acceleration = -m * (force + gamma_friction_driving_damping * v)

    return acceleration
```

Figure 24: Example Prompt in Oscillator1 - For Meta Strategy Generator.

```
ID Score      : -1.16205 (in-distribution performance)
OOD Score     : -1.00194 (out-of-distribution performance)
Eval Time     : 0.698804s (computational efficiency)
Avg Accuracy  : -1.082 (overall accuracy)
Gen Gap       : 0.160103 (generalization stability)
Efficiency    : 0.58865 (time-normalized performance)

**In-Distribution (ID) Dataset Analysis:**
  Worst Region Statistics:
    - Sample Count: 520 samples
    - Threshold: 0.101862
    - Mean Residual: 0.0907549
    - Mean Abs Residual: 0.109957
    - Max Abs Residual: 0.113557
    - Std Residual: 0.0622102
  Input Variables in Worst Region:
```

```
    x0: [-0.512046, 0.444764]     x1: [0.11177, 0.212819]    Target Variable
(y) in Worst Region:
    y: [-0.104672, 0.110646]
**Out-of-Distribution (OOD) Dataset Analysis:**
  Worst Region Statistics:
    - Sample Count: 480 samples
    - Threshold: 0.0978116
    - Mean Residual: 0.0446015
    - Mean Abs Residual: 0.103459
    - Max Abs Residual: 0.111418
    - Std Residual: 0.0934521
  Input Variables in Worst Region:
    x0: [-0.562765, 0.495206]     x1: [0.0432888, 0.221801]    Target
Variable (y) in Worst Region:
    y: [-0.103036, 0.10915]
```

Figure 25: Example Prompt in Oscillator1 - For Meta Strategy Generator.

```
## Available Analysis Tools
ccm_causality, corr, corr_dim_relation, dtw_distance, fft_cross_freq,
granger_causality, ks_test_diff, lin_reg, lyapunov_relation, mutual_info,
mutual_info_regression_score, pca_mapping, pearson_corr, residual_var,
spearman_corr, wavelet_corr
## Pattern Analysis Instructions
1. **Examine Mathematical Structures**: Look across all equations for:
   - Recurring mathematical functions (trigonometric, exponential,
polynomial)
   - Similar variable interaction patterns (x*y, x+y, x^n, etc.)
   - Consistent coefficient magnitudes or sign patterns
   - Common denominators or fraction structures
   - Similar nesting or grouping of terms

2. **Identify Shared Elements**: Find mathematical components that appear
in multiple high-performing equations

3. **Interpret Mathematical Significance**: Explain what these patterns
might indicate about:
   - Underlying physical relationships
   - Mathematical constraints or principles
   - Optimal solution characteristics

4. **Generate Guidance**: Create a detailed prompt that will guide future
equation generation based on successful patterns

5. **Select Analysis Tools**: Choose up to 5 data analysis tools that could
help validate the patterns you identified

## RESPONSE FORMAT
Return ONLY JSON with keys: structure_insight_prompt_for_generator_LLM,
analysis_tools.
- structure_insight_prompt_for_generator_LLM: Detailed guidance based on
your pattern analysis for generating better equations
- analysis_tools: Array of up to 5 tool names from the available list

Focus on mathematical commonalities and their significance in equation
generation.
```

Figure 26: Example Prompt in Oscillator1 - For Meta Strategy Generator.

**Partial Data Analysis Tools Usage in CRK36:**

**pearson_corr:**

```
## Correlation Analysis: x0_y
 **Strong correlation detected** (r=1.000)
Consider incorporating a strong relationship between x0_y and output in
your equation.
The relationship appears to be positive - as one increases, the other
increases.
```

**spearman_corr:**

```
## Correlation Analysis: x0_y
 **Strong correlation detected** (r=1.000)
Consider incorporating a strong relationship between x0_y and output in
your equation.
The relationship appears to be positive - as one increases, the other
increases.
```

**lin_reg:**

```
## Linear Regression Analysis: x1_y
 **Strong linear relationship** (R²=0.996)
Linear equation: output ≈ 0.000 × x1_y + 0.000
Consider incorporating linear terms in your equation.
```

**residual_var:**

```
## Residual Variance Analysis: x0_y
**Residual variance**: 0.057119
Low residual variance suggests the linear relationship captures most
variation.
```

**mutual_info:**

```
## Mutual Information Regression Analysis: x1_y
**Mutual information score**: 8.294050
Mutual information regression measures the dependency between variables in
a regression context.
```

Figure 27: Examples of Data Analysis Tools Usage.

