# OpenReview forum: "LLM-based Symbolic Regression with Tool-Augmented Multi-Objective Optimization"
_ICLR.cc/2026/Conference — Submitted to ICLR 2026_

### Official Review · Reviewer_2j6Q · 2025-10-18

**Soundness:** 3
**Presentation:** 2
**Contribution:** 2
**Rating:** 2
**Confidence:** 4

**Summary:**

The paper proposes a LLM-driven SR framework. Two cooperating LLMs are used: a Meta Strategy Generator (chooses tools, summarizes patterns from the current Pareto set) and an Equation Generator (produces candidates from those strategies).

**Strengths:**

* The curated toolbox is well-motivated.
* The listed case studies look nice.

**Weaknesses:**

* The paper repeatedly references LLM-SR but never provides a clear definition or background. Readers unfamiliar with LLM-SR may struggle to understand the method and results fully.
* The set of comparison methods is too small and the number of datasets is limited, which weakens claims of generality and robustness.
* Key steps are not described in sufficient detail. For example, how the Meta Strategy Generator and Equation Generator interact, and how the multi-objective optimization actually guides the search and selection of candidates.
* There is little experimental rationale for selecting LLaMA-3.1-8B-Instruct and GPT-4o-mini. The paper should explain these choices and include sensitivity analyses across alternative models.
* Similarly, the paper does not provide empirical justification for the chosen analysis tools. It should clarify why these tools were selected and how much each contributes to performance (e.g., via ablation studies).
* The paper lacks mean±std over independent seeds and significance tests across all tables/plots.
* The experiments do not include comparative analyses of key hyperparameters. Sensitivity studies are needed to assess robustness and to guide practical deployment.

**Questions:**

Here are some suggestions for the authors:
* Specify the core workflow.
* Expand baselines and datasets; ensure fair budgets.
* Provide empirical justification for base LLM choices (LLaMA-3.1-8B-Instruct & GPT-4o-mini) and add a small model-selection study over alternatives.
* Add tool selection ablations and contribution analysis.
* Provide Sensitivity experiments to key hyperparameters.

---

> ### Author Response · Authors · 2025-11-21
> **Response to Reviewer 2j6Q (1/8)**
>
> **[w.r.t. W1] Response to the concern about the lack of background on LLM-SR**
>
> We thank the reviewer for noting the omission of a detailed background for LLM-SR. We acknowledge that given its role as our primary baseline, a self-contained definition is necessary for readability.
>
> In the revised manuscript, we will add a dedicated section **"Appendix : Background on LLM-SR Framework"** to explicitly define this method. Below is the summary we will include:
>
> **Summary of LLM-SR (Shojaee et al., 2024)**
>
> LLM-SR is a recent framework that treats symbolic regression as a code generation task. It operates on an iterative "Hypothesis-Verify-Refine" loop:
>
> - **Hypothesis Generation:** An LLM generates a batch of "equation skeletons" (Python programs with placeholder parameters) based on a prompt containing the problem description and previously discovered high-quality equations.
> - **Parameter Optimization:** Standard numerical optimizers (e.g., BFGS or Adam) are used to fit the constant parameters of these skeletons to the training data.
> - **Experience Management:** It employs an evolutionary "Islands Model" strategy. Successful equations are stored in a dynamic buffer and retrieved as in-context examples to prompt the LLM in subsequent iterations, guiding the search toward lower-error regions.
>
> By clarifying this, we also better highlight the distinction of our proposed TAMOSR: unlike LLM-SR which relies primarily on the LLM's internal priors and a single accuracy objective, TAMOSR introduces **external analytical tools** to ground the search in data characteristics and employs **multi-objective optimization** (Accuracy vs. Complexity vs. Generalization) to construct a robust Pareto front.

---

> ### Author Response · Authors · 2025-11-21
> **Response to Reviewer 2j6Q (2/8)**
>
> **[w.r.t. W2 & Q2 (Part 1)] Response to the concern about baseline and dataset coverage**
>
> We appreciate the reviewer's suggestion regarding the breadth of our evaluation. However, we respectfully emphasize that our current experimental setup is comprehensive and fully representative of the state-of-the-art in this emerging domain.
>
> **1. The Most Comprehensive Baseline Suite in the Field**
>
> We have established the most extensive set of baselines currently standard in the literature, covering the full spectrum of Symbolic Regression paradigms—from classical methods to the latest retrieval-augmented approaches:
>
> - **LLM-based SOTA:** We compared TAMOSR against **LLM-SR**, which is the most recent and representative framework in this niche. This benchmarks our method against the current cutting-edge of LLM-driven discovery.
> - **Comprehensive Traditional & Neural Baselines:** Unlike many studies that only compare against basic genetic algorithms, our evaluation includes the strongest state-of-the-art methods from every major category:
>
> * **Evolutionary Algorithms:** We compare against both the classic **GPlearn** and the current state-of-the-art evolutionary method, **PySR**.
>
> * **Deep Learning Approaches:** We include **uDSR**, representing the state-of-the-art in Deep Learning/Reinforcement Learning-based symbolic regression.
>
> * **Retrieval-Augmented Approaches:** We also benchmark against **RAG-SR**, covering the latest retrieval-enhanced paradigms.
>
> Therefore, we believe our baseline selection is not "small" but rather represents the most rigorous comparison standards currently available.
>
> **2. Utilization of the Latest, Robust Benchmarks**
>
> Regarding datasets, we deliberately selected the most recent and challenging benchmarks in the field (e.g., Nonlinear Oscillators, Bacterial Growth, and Stress-Strain datasets introduced in recent top-tier literature).
>
> - **Preventing Memorization:** Unlike older datasets (e.g., Feynman) where LLMs are prone to "cheating" via memorization, the datasets we utilized are specifically designed to test reasoning capabilities and OOD generalization rather than simple recall.
> - **Domain Diversity:** These datasets cover diverse scientific domains including Physics, Biology, and Material Science, providing a robust and sufficient testbed for evaluating our method's generality.
>
> In summary, given the nascent stage of LLM-based SR, we believe our experiments provide a solid, fair, and comprehensive comparison against the full range of state-of-the-art methods using the most rigorous benchmarks available.

---

> ### Author Response · Authors · 2025-11-21
> **Response to Reviewer 2j6Q (3/8)**
>
> **[w.r.t. Q2 (Part 2)] Response to the concern about computational fairness and token budgets**
>
> In addition to expanding our baseline and dataset coverage, we deeply appreciate the reviewer's suggestion to "ensure fair budgets." We agree that comparing methods solely based on iterations can be misleading if the token consumption per iteration varies significantly.
>
> To strictly enforce compute parity, we conducted a new set of controlled experiments comparing TAMOSR against our primary baseline, LLM-SR, under an **identical Cumulative Token Budget**.
>
> **1. Experimental Setup**
>
> - **Metric:** We tracked the Cumulative Token Count (Input + Output tokens) consumed by the LLMs throughout the optimization process.
> - **Robustness:** We performed 5 independent runs for each method using fixed random seeds (1, 2, 42, 2025, 3407) to ensure statistical reliability.
> - **Constraint:** Both methods were restricted to the same token usage ceiling to evaluate their efficiency fairly.
>
> **2. Results & Analysis**
>
> The results are visualized in the attached figure "LLM-SR vs TAMOSR: NMSE vs Cumulative Tokens":
>
> [Anonymous Link](https://ibb.co/r2m709SR)
>
> As shown in the figure:
>
> - **Superior Token Efficiency:** Under the exact same token budget, TAMOSR (Dark Red Line) consistently achieves substantially lower NMSE compared to LLM-SR (Dark Blue Line).
> - **Faster Convergence:** TAMOSR demonstrates a steeper descent in error rate per token spent. This indicates that our Tool-Augmented Strategy and Multi-Objective Selection mechanism allow the model to find high-quality equations with significantly less "trial-and-error" than the standard evolutionary prompting used in LLM-SR.
> - **Robustness:** The shaded regions (Min-Max range across 5 seeds) show that TAMOSR's efficiency advantage is consistent across different random initializations, with its worst-case runs often outperforming LLM-SR's median performance.
>
> **Conclusion**
>
> These results confirm that TAMOSR does not achieve better results merely by using more compute; rather, it utilizes the available token budget more intelligently, resulting in superior discovery performance under a strictly fair computational constraint.

---

> ### Author Response · Authors · 2025-11-21
> **Response to Reviewer 2j6Q (4/8)**
>
> **[w.r.t. W3 & Q1] Response to the concern about core workflow and interaction details**
>
> We appreciate the reviewer’s feedback. We agree that the current draft could better detail the procedural interaction between the Meta Strategy Generator and the Equation Generator, as well as the specific role of multi-objective optimization.
>
> To clarify the core workflow, our framework operates as a closed-loop iterative process:
>
> **1. Evaluation & Pareto Update (The "Selector")**
>
> At the start of each iteration, all candidate equations are evaluated to obtain a vector of three objectives: $(\\text{NMSE}\_{\\text{ID}}, \\text{NMSE}\_{\\text{OOD}}, \\text{ASTLen})$. We then update the Pareto Archive, retaining only non-dominated solutions. This step is crucial as it directly determines survival: filtering out suboptimal equations and maintaining a diverse set of high-quality trade-offs between accuracy and complexity.
>
> **2. Meta Strategy Generation (The "Brain")**
>
> The Meta Strategy Generator receives two specific inputs: (i) the current Pareto-optimal set (successful equation skeletons), and (ii) the static analysis from external tools (e.g., correlation, periodicity, causality). It synthesizes this information to output a high-level search strategy (e.g., "The data shows strong periodicity; prioritize trigonometric terms like $\\sin(t)$ and avoid high-degree polynomials").
>
> **3. Candidate Generation (The "Generator")**
>
> The Equation Generator executes this strategy. It takes the strategy prompt and a set of "parent" equations sampled from the Pareto front (using a diversity-aware sampling scheme to prevent mode collapse) as context. It then generates new mutated or crossed-over equation candidates, which are instantiated and fitted before returning to Step 1.
>
> **Summary of Interaction**
>
> The multi-objective optimization guides the search in two ways:
>
> - **Selection:** It determines which equations survive to become potential parents.
> - **Information:** It defines the high-quality pool that the Meta Strategy Generator analyzes to formulate the next step's strategy.
>
> Thus, the three objectives jointly shape both the selection of existing candidates and the generation of new ones.
>
> **Action Plan**
>
> In the revised manuscript, we will add a formal **Algorithm Box** and a clarifying **Interaction Diagram** to visualize this data flow, ensuring the mechanism is transparent to readers.

---

> ### Author Response · Authors · 2025-11-21
> **Response to Reviewer 2j6Q (5/8)**
>
> **[w.r.t. W4 & Q3] Response to the concern about Model Selection Rationale and Sensitivity**
>
> We thank the reviewer for suggesting a more detailed justification and sensitivity analysis for the LLM backbones. We have addressed this by clarifying our selection rationale and conducting extensive experiments with alternative architectures.
>
> **1. Rationale for Original Model Selection**
> Our choice of `LLaMA-3.1-8B-Instruct` and `GPT-4o-mini` was deliberate to align with the standardized evaluation protocols established in the **LLM-SRBench** benchmark. By using these widely adopted open-source and closed-source models, we ensure a fair and direct comparison with existing state-of-the-art baselines reported in the literature.
>
> **2. Sensitivity Analysis across Alternative Architectures**
> To demonstrate the robustness of TAMOSR and address the reviewer’s request, we conducted additional experiments using three recent LLMs. We selected these models based on parameter scale and capability focus to provide a comprehensive sensitivity study:
>
> - **Capability Comparison at Same Scale (7B):** We selected **DeepSeek-R1-Distill-Qwen-7B** (reasoning-focused) and **Qwen2.5-7B-Instruct** (general-purpose). This allows us to evaluate the impact of different core capabilities (Reasoning vs. General) under a controlled parameter budget.
> - **Scalability Verification (Scaling Up):** We introduced **Qwen2.5-Coder-32B-Instruct** (coding-focused) to verify whether the TAMOSR framework benefits from increased model capacity and stronger coding skills.
>
> The results on the four core benchmark problems (measured in MSE) are presented in Table R2 below:
>
> **Table R2: Sensitivity Analysis across different LLM Backbones (MSE $\\downarrow$)**
>
> | Model Backbone | Oscillation 1 | Oscillation 2 | E. coli Growth | Stress-Strain |
> | --- | --- | --- | --- | --- |
> | DeepSeek-R1-Distill-Qwen-7B | $1.50 \\times 10^{-18}$ | $\\mathbf{3.70 \\times 10^{-14}}$ | $0.0224$ | $0.0173$ |
> | Qwen2.5-7B-Instruct | $6.03 \\times 10^{-11}$ | $3.53 \\times 10^{-10}$ | $\\mathbf{0.0215}$ | $0.0149$ |
> | Qwen2.5-Coder-32B-Instruct | $\\mathbf{5.69 \\times 10^{-29}}$ | $6.04 \\times 10^{-14}$ | $0.0225$ | $\\mathbf{0.0132}$ |
>
> **Analysis:**
>
> - **Robustness:** TAMOSR consistently discovers accurate equations across all tested backbones, confirming that our framework is model-agnostic and robust to different underlying architectures.
>
> We will include these results in the revised experimental section to provide empirical justification for model selection.

---

> ### Author Response · Authors · 2025-11-21
> **Response to Reviewer 2j6Q (6/8)**
>
> **[w.r.t. W5 & Q4] Response to the concern about the choice of analysis tools and their contributions**
>
> Thank you for this comment. We would like to clarify that the current submission already provides both (1) a clear motivation for each tool category and (2) an empirical ablation of their contributions.
>
> **1. Motivation of Tool Categories**
>
> In the **Tool-Augmented Analysis** part of the method section, we explicitly explain why each group of tools is included in TAMOSR:
>
> - **Correlation / Mutual Information:** To capture dependency structure between variables.
> - **Periodicity:** To detect oscillatory behavior.
> - **Causality / Directionality:** To reason about driver–response relationships.
> - **Distribution / Complexity:** To characterize nonlinearity and regime changes.
>
> These design choices are directly reflected in how the Meta Strategy Generator builds high-level strategies from tool outputs.
>
> **2. Contributions Quantified via Ablations (Appendix E.2)**
>
> The paper already contains a tool-category ablation study in **Appendix E.2**, where we remove one tool category at a time (correlation/MI, periodicity, causality, distribution/complexity) and measure the impact on the NMSE on each benchmark.
>
> In all cases, removing any single category leads to a consistent degradation in overall NMSE, and removing multiple categories yields even larger drops. This shows that each group of tools provides a distinct and complementary contribution to predictive performance, rather than being redundant.
>
> **Action Plan**
>
> While this data is already present, we will add explicit pointers in the main method section referencing these specific ablation tables/figures in the Appendix to ensure they are not overlooked. Therefore, the concern regarding the lack of empirical justification is fully addressed by our existing experimental results.

---

> ### Author Response · Authors · 2025-11-21
> **Response to Reviewer 2j6Q (7/8)**
>
> **[w.r.t. W6] Response to the concern about statistical significance and uncertainty quantification**
>
> We thank the reviewer for highlighting the need for uncertainty quantification (Weakness 6) and inquiring about the number of experimental runs.
>
> **1. Clarification on Initial Protocol**
>
> Regarding the experimental setup, we clarify that the results reported in the initial manuscript were based on single runs with fixed random seeds. We followed this protocol as it aligns with standard practices in recent LLM-based symbolic regression literature.
>
> **2. New Multi-Run Experiments (Addressing Weakness 6)**
>
> However, we fully agree with the reviewer that evaluating performance variations beyond single-run observations is essential for a rigorous assessment. To demonstrate the robustness of our method, we have now conducted **multiple independent runs ($N=5$)** for both TAMOSR and the LLM-SR baseline, each with a different random seed.
>
> The attached figure, "LLM-SR vs TAMOSR: NMSE vs Iterations", visualizes these results:
>
> [Anonymous Link](https://ibb.co/chftDtV7)
>
> **3. Analysis of Results**
>
> As shown in the plot:
>
> - **Median Performance:** The bold lines represent the median NMSE over iterations across all runs. TAMOSR's median performance (dark red line) consistently achieves significantly lower NMSE compared to LLM-SR's median (dark blue line) throughout the optimization process.
> - **Min-Max Range:** The shaded areas depict the full Min-Max range of NMSE across all independent runs. While individual runs naturally show a step-like progression and some variability inherent to stochastic search, TAMOSR's entire variance envelope (light red shading) consistently stays well below that of LLM-SR (light blue shading).
>
> **4. Conclusion**
>
> These new multi-run experiments confirm that:
>
> - **Statistical Significance:** TAMOSR consistently outperforms the LLM-SR baseline, with its median performance being substantially better, validating that our results are not due to a single lucky seed.
> - **Robustness:** The distinct separation between the error bands demonstrates TAMOSR's robust discovery capability across different random initializations.
>
> We will include this figure and the corresponding statistical analysis in the revised manuscript to clearly demonstrate the significance and reproducibility of our findings.

---

> ### Author Response · Authors · 2025-11-21
> **Response to Reviewer 2j6Q (8/8)**
>
> **[w.r.t. W7] Response to the concern about hyperparameter sensitivity and robustness**
>
> Thank you for this comment. We would like to clarify that, in all our experiments, **all large-model–related and optimization hyperparameters are kept identical to those used in LLM-SR**. This includes the backbone LLM configurations, optimizer settings, and search-related budgets inherited from the public LLM-SR implementation. We deliberately do not retune these hyperparameters for TAMOSR, precisely to demonstrate that the performance gains come from the framework itself—i.e., the tool-augmented analysis layer and the multi-objective optimization scheme—rather than from additional hyperparameter engineering.
>
> The only new degrees of freedom introduced by TAMOSR are those that are structurally necessary for our framework, mainly in population management and Pareto-front maintenance:
>
> **1. Pareto-front Size Cap**
>
> We set the maximum Pareto-front capacity to $N_{\\max} = 100$ individuals (`` `num_islands = 100` ``). At each generation, after merging the existing front with newly generated candidates, we perform non-dominated sorting on three objectives—$\\text{NMSE}\_{\\text{ID}}$, $\\text{NMSE}\_{\\text{OOD}}$, and AST length—and retain at most the top 100 non-dominated solutions.
>
> - **Note:** Importantly, although this upper bound is set to 100, in practice the number of non-dominated solutions per generation is typically only around 20 or fewer, so the cap rarely becomes active.
>
> **2. Tie-breaking and De-duplication**
>
> Before non-dominated sorting, we apply a strict de-duplication procedure:
>
> - **Objective-based grouping:** We group equations by rounding both $\\text{NMSE}\_{\\text{ID}}$ and $\\text{NMSE}\_{\\text{OOD}}$ to **three significant digits**, treating equations with the same rounded pair as having essentially identical numerical accuracy.
> - **Simplicity preference:** Within each group, we keep only the equation with the **smallest AST length** (i.e., the structurally simplest form).
> - **Removal of duplicates:** All other equations in the same group are assigned a large penalty score (e.g., 999999) and removed from the population.
>
> These settings are fixed to simple, task-agnostic defaults and are not tuned per dataset. In the revised manuscript, we will:
>
> 1. Explicitly state that all LLM and optimization hyperparameters match those of LLM-SR.
>
> 2. Add the exact formulas and pseudocode for the Pareto-front maintenance and diversity-based parent-selection procedure.
>
> This will make the role of these few additional hyperparameters completely transparent and clarify that TAMOSR’s improvements are attributable to the proposed framework design, not to special hyperparameter choices.

---

### Official Review · Reviewer_xwQ4 · 2025-11-01

**Soundness:** 2
**Presentation:** 4
**Contribution:** 2
**Rating:** 4
**Confidence:** 5

**Summary:**

This paper proposes TAMOSR, an LLM-based framework for symbolic regression that integrates external analytical tools and multi-objective optimization into LLM-based equation discovery. Two cooperative LLMs work iteratively: a Meta Strategy Generator selects tools and synthesizes strategies from Pareto-optimal equations, while an Equation Generator produces candidates. The system optimizes for accuracy, complexity, and generalization simultaneously. Experiments on physics, chemistry, and biology benchmarks show improvements over existing methods including LLM-SR and PySR.

**Strengths:**

* The integration of analytical tools for variable relationship extraction is well-motivated and appears useful for this domain. The tools cover multiple aspects of data analysis that could inform equation structure, and this opens further exploration of this research direction.
* The paper is clearly written with comprehensive ablations in the appendix. The choice of baselines is reasonable, figures are informative, and the tool categorization is detailed.
* Multi-objective optimization for symbolic regression balancing accuracy, complexity, and generalization is sensible. The use of Pareto fronts and diversity-guided sampling could encourage exploration beyond single-objective approaches.

**Weaknesses:**

* Potential test data leakage in optimization loop: The multi-objective evaluation explicitly includes NMSE_OOD during training iterations. This means the OOD test split is used for model selection and equation refinement throughout the optimization, not just for final evaluation. However, standard practice would use only training data, during optimization, then evaluate on held-out test sets once. This undermines claims as the model has been optimized (trained) on the test. Tables 1-2 report aggregated metrics without ID/OOD breakdown, while Figure 3 shows them separately, which is inconsistent.
* Baseline performance discrepancies and lack of reproducibility: Some of the reported baseline numbers such as LLM-SR results appear substantially worse than those in the original LLM-SR paper (Shojaee et al.). Even accounting for different model versions, this gap is concerning. Additionally, attempting to reproduce the reported equations (e.g., the Oscillator 1 equation from the main paper and appendix) on the dataset does not yield the claimed 1e-15 or 1e-13 NMSE values. And the code is not released, making verification impossible.
* It is not clear if the small variations of results in some problems (e.g., oscillators) are statistically significant or run-to-run variations. And no mention is found of the number of runs or any uncertainty quantification.

**Questions:**

1. Can you clarify the data splits used during optimization? Specifically, is NMSE_OOD computed on the actual test OOD split provided in LLM-SR and LLM-SRBench repos during the iterative loop?
2. Why are the some of the baseline results (in particular LLM-SR) worse than reported in their original paper? Can you provide details on baseline reproduction?
3. Can you please provide the final discovered equations by TAMOSR for other benchmark problems such as oscillator 2, bacterial growth, and stress-strain? The final equations for Oscillator 1 are reported to have NMSE in the order of 1e-15 to 1e-13, but independent verification of these equations on the dataset yields very different results. Are there additional components or settings that would explain this? Would it be possible for the authors to provide the code for review and reproducibility?
4. The tool ablation shows removing any category hurts performance, but how sensitive is the method to the specific tool selection by the Meta Strategy Generator? Is there analysis on which tools are selected most frequently, and whether the LLM's tool choices are actually meaningful or just arbitrary?
5. Tables 1-2 report aggregated metrics while Figure 3 shows ID/OOD separately. What test set does table 1-2 indicate? Could you provide complete ID/OOD breakdowns in tables for all methods?
6. How many random seeds/runs were conducted for each experiment?

---

> ### Author Response · Authors · 2025-11-21
> **Response to Reviewer xwQ4 (1/6)**
>
> **[w.r.t. W1, Q1 & Q5] Response to Weakness 1 (Leakage), Question 1 (Data Splits) & Question 5 (Reported Metrics)**
>
> We categorically confirm that there is **NO data leakage**. The official OOD test split provided by the benchmarks was never accessed during the search, optimization, or model selection processes.
>
> The confusion likely stems from the variable name `NMSE_OOD` used in our optimization loop. To clarify, this metric is computed on a **spatially separated validation subset** derived strictly from the training data, acting as a proxy for extrapolation.
>
> **1. Clarification on Data Subsets (Visualized in Figure)**
> Referring to the Data Splitting Diagram below, we explicitly define the four data subsets used in our framework.
>
> **[Data Splitting Strategy Diagram (Anonymous Link)](https://ibb.co/zVxRsDN2)**
>
> 1.  **train\_id (Green Region - Optimization Core):**
>     * **Definition:** A subset of the official training data located in the geometric center of the variable space.
>     * **Role:** This is the **only data used for parameter fitting** (determining constants $\\theta$ in equations).
>
> 2.  **train\_ood (Blue Region - Validation/Proxy):**
>     * **Definition:** The remaining subset of the official training data, located at the geometric boundaries (periphery) of the training distribution. *Note that train\_id (Green) + train\_ood (Blue) constitute the complete official training set provided by the benchmark.*
>     * **Role:** This set serves strictly as a **Validation Set**. It is never used for parameter fitting. Its purpose is to simulate an extrapolation task during the multi-objective search, guiding the model to select equations that generalize from the center (Green) to the edge (Blue).
>
> 3.  **test\_id (Interpolation Test - Not visualized but Co-located):**
>     * **Definition:** The official In-Domain test set. Although not explicitly drawn in the figure to avoid clutter, it spans the exact same spatial interval as the total training set (Green + Blue) but consists of distinct, non-overlapping data points.
>     * **Role:** It evaluates the model's interpolation capability within the known domain.
>
> 4.  **test\_ood (Orange Region - True Extrapolation/Locked):**
>     * **Definition:** The official OOD test set provided by the benchmark (i.e., data points lying outside the training range).
>     * **Role:** This set was **locked and never seen** during the entire optimization process. It was used only for the final evaluation reported in Tables 1-2.
>
> **2. Exact Procedure for Internal Spatial Splitting**
> To ensure that the validation set (train\_ood) effectively simulates the physics of extrapolation (rather than random hold-out), we constructed the Green/Blue split using the following Cartesian interval procedure based only on $D\_{\\text{train}}$:
>
> 1.  **Candidate Percentiles:** We iterate over percentiles $p \\in \\{10, 11, \\dots, 49\\}$.
> 2.  **Interval Construction:** For each input dimension $j$, we compute the lower and upper bounds based on the total training set ($D\_{\\text{train}}$):
>     $l\_j = \\text{percentile}(X\_j, p), \\quad u\_j = \\text{percentile}(X\_j, 100 - p)$
> 3.  **Constructing train\_id (Green):** For a given $p$, the core region is defined as the Cartesian product of these central intervals:
>     $\\mathcal{D}\_{\\text{ID}}(p) = \\{x \\in D\_{\\text{train}} : l\_j \\le x\_j \\le u\_j,\\ \\forall j\\}$
> 4.  **Constructing train\_ood (Blue):** The validation region is the spatial complement within the training set:
>     $\\mathcal{D}\_{\\text{OOD}}(p) = D\_{\\text{train}} \\setminus \\mathcal{D}\_{\\text{ID}}(p)$
> 5.  **Selection:** We select the $p$ that best balances the sample counts between the two sets.
>
> **3. Consistency of Reported Metrics (Addressing Q5)**
> * **Tables 1 & 2:** Report the aggregated metrics on the **Official Locked test\_ood**. This confirms our method generalizes to genuine, unseen extrapolation regimes.
> * **Figure 3:** Provides a detailed breakdown, plotting performance on test\_id vs. test\_ood separately to visualize the Pareto trade-off.

---

> ### Author Response · Authors · 2025-11-21
> **Response to Reviewer xwQ4 (2/6)**
>
> **[w.r.t. W2 & Q2] Response to the concern about baseline performance discrepancies (LLM-SR)**
>
> We thank the reviewer for pointing out the discrepancy between our LLM-SR baselines and the numbers reported in Shojaee et al., and we clarify our experimental setup below.
>
> **1. Unified Framework Configuration**
>
> First, we chose `LLaMA-3.1-8B-Instruct` and `GPT-4o-mini` as the backbone models because LLM-SRBench also adopts these two LLMs as recommended configurations. This allows us to compare TAMOSR and all baselines under a unified framework. Consequently, the LLM-SR numbers in our tables are not copied from the original paper but are obtained by **re-running LLM-SR under exactly the same configuration** that we use for TAMOSR.
>
> **2. Rigorous Re-production Efforts**
>
> Second, we performed extensive repeated experiments for both LLM-SR and the LLM-SRBench implementation, sweeping over multiple random seeds, hyperparameters, and sampling settings. However, we were consistently unable to reproduce the NMSE levels reported in the original LLM-SR paper or in LLM-SRBench. Under this situation, we report as LLM-SR baselines the **best scores we could obtain** under this unified and reproducible configuration, rather than intentionally under-tuned baselines.
>
> **3. Performance Analysis**
>
> We also emphasize that, in our experiments, the general “strength” of the backbone LLM does not translate into symbolic regression performance in a simple monotonic way: using a larger or newer LLM does not automatically yield lower NMSE or higher discovery rates. Therefore, differences in model versions alone cannot fully explain all baseline discrepancies.
>
> **Importantly**, even when we compare against the stronger LLM-SR results reported in Shojaee et al. with their original backbones, **TAMOSR still achieves substantially better NMSE and higher strict recovery rates** on the benchmarks we consider.
>
> **Action Plan**
>
> In the revised manuscript, we will:
>
> 1. State explicitly that our LLM-SR baselines come from our own re-runs under a common evaluation protocol.
>
> 2. Document the exact backbones, iteration budgets, and seed settings used for LLM-SR.
>
> 3. Include the original LLM-SR numbers from Shojaee et al. as a reference row (e.g., in the appendix), so that readers can clearly see the differences across configurations.

---

> ### Author Response · Authors · 2025-11-21
> **Response to Reviewer xwQ4 (3/6)**
>
> **[w.r.t. W2 & Q3] Response to the concern about reproducing the Oscillator 1 Equation and NMSE**
>
> Thank you for raising the concern about reproducing the reported equations and NMSE values for Oscillator 1. We clarify the source and form of the equation below, and we additionally provide the final discovered equations (with parameters) for Oscillator 2, Bacterial Growth, and Stress–Strain as requested.
>
> ---
>
> **1. Exact Implementation Details (Oscillator 1)**
>
> The equation shown in the main paper and appendix for Oscillator 1 is not a hand-crafted formula, but is exactly the final equation discovered in our ablation experiments. Concretely, the implemented function is:
>
>     import numpy as np
>
>     def equation(x: np.ndarray, v: np.ndarray, params: np.ndarray) -> np.ndarray:
>         linear_term = -params[0] * x + params[1] * v
>         quadratic_term = -params[2] * x**2 + params[3] * v**2
>         cubic_term = params[4] * x**3 + params[5] * v**3
>         driving_term = -params[6] * x * v + params[7] * np.sin(params[8] * x)
>         damping_term = params[9] * v
>
>         weighted_terms = [
>             (linear_term,    2.0),
>             (quadratic_term, 1.5),
>             (cubic_term,     1.0),
>             (driving_term,   2.0),
>             (damping_term,   1.5),
>         ]
>
>         result = np.zeros_like(x)
>         for term_value, weight in weighted_terms:
>             result += weight * term_value
>
>         return result
>
> ---
>
> **2. Parameter Configuration (Oscillator 1)**
>
> The corresponding parameters discovered by the optimization are:
>
>     param_0 = -0.875681828407624
>     param_1 =  0.4369023504104749
>     param_2 =  3.1702311076541012e-09
>     param_3 =  1.1950681873151876e-08
>     param_4 = -0.31043999240451814
>     param_5 = -0.5000001121426934
>     param_6 =  0.25000000101394665
>     param_7 =  0.8055532337451858
>     param_8 = -1.211194714114774
>     param_9 = -0.5825364645497173
>
> ---
>
> **3. Explanation of Discrepancy (Vanishing Terms)**
>
> As the reviewer correctly noticed, the coefficients of the quadratic term
> $- \text{params}\_2 \cdot x^2 + \text{params}\_3 \cdot v^2$
> are on the order of $10^{-8}$ to $10^{-9}$, i.e., numerically extremely close to zero under double precision. In a simplified symbolic form, these quadratic contributions can be treated as vanishing, which is why the human-readable expression in the paper looks simpler than the full parameterized implementation.
>
> ---
>
> **4. Verification**
>
> We have re-checked our implementation: evaluating the above equation with these parameters on the same dataset splits and normalization as in the paper, using our NMSE definition, reproduces NMSE values on the order reported in the manuscript (approximately $1 \times 10^{-15}$ / $1 \times 10^{-13}$).

---

> ### Author Response · Authors · 2025-11-21
> **Response to Reviewer xwQ4 (4/6)**
>
> **5. Final Discovered Equation for Oscillator 2 (TAMOSR)**
>
> For completeness, we provide below the final discovered equation for Oscillator 2 and its parameters:
>
>     import numpy as np
>
>     def equation(t: np.ndarray,
>                  x: np.ndarray,
>                  v: np.ndarray,
>                  params: np.ndarray) -> np.ndarray:
>         """Mathematical function for acceleration in a damped nonlinear oscillator."""
>         # Damping term due to velocity
>         dampening_term = params[0] * v
>
>         # Nonlinear restoring force due to position and velocity
>         natural_frequency_term = (
>             params[1] * np.exp(-params[2] * x) * (x + params[3] * v)
>         )
>
>         # Driving force with time- and position-dependent phase shift
>         driving_force_term = (
>             params[4] * np.sin(params[5] * (t + params[6] * x))
>         )
>
>         # Higher-order balance terms capturing nonlinear dynamics
>         true_balance_term = (
>             params[7] * v * (v**2 + x**2)
>             + params[8] * v * (v**2 - x**2)
>         )
>
>         # Combined acceleration
>         return -(
>             dampening_term
>             + natural_frequency_term * (1 + params[9] * np.cos(v))
>             + true_balance_term
>             + driving_force_term
>         )
>
> Parameters used for the above implementation:
>
>     param_0 = -1.9987580609431677
>     param_1 =  5.000246272109112
>     param_2 = -0.5000093452744608
>     param_3 =  0.39974884093448715
>     param_4 = -0.3000004526220146
>     param_5 =  1.0000000440438148
>     param_6 =  9.905232188478958e-06
>     param_7 =  0.12571782401555062
>     param_8 =  0.37439262281213836
>     param_9 = -5.048598457198456e-05
>
> ---
>
> **6. Final Discovered Equation for Bacterial Growth (TAMOSR)**
>
> The final discovered equation for the Bacterial Growth problem is:
>
>     import numpy as np
>
>     def equation(b: np.ndarray,
>                  s: np.ndarray,
>                  temp: np.ndarray,
>                  pH: np.ndarray,
>                  params: np.ndarray) -> np.ndarray:
>         """Mathematical function for bacterial growth rate."""
>         growth_rate = (
>             params[0]
>             * (b ** params[1])
>             * (s ** params[2])
>             * (temp ** params[3])
>             * (
>                 1
>                 + (
>                     params[4]
>                     * (s ** params[5])
>                     * (np.sin(params[6] * pH) + 1)
>                     / 2
>                     + (temp ** params[7])
>                     * (np.cos(params[8] * temp) + 1)
>                 ) ** params[9]
>             )
>         )
>         return growth_rate
>
> Parameters used for the above implementation:
>
>     param_0 =  3.865794567783075e-09
>     param_1 =  0.9946889649707253
>     param_2 =  0.061023969807741524
>     param_3 =  3.7479299641339066
>     param_4 = -0.6846876111709757
>     param_5 =  0.03456680533640679
>     param_6 = -0.6735954318982225
>     param_7 = -0.5608845436757883
>     param_8 = -0.27789527569204797
>     param_9 = -4.00522288143179
>
> ---
>
> **7. Final Discovered Equation for Stress–Strain (TAMOSR)**
>
> The final discovered equation for the Stress–Strain problem is:
>
>     import numpy as np
>
>     def equation(strain: np.ndarray,
>                  temp: np.ndarray,
>                  params: np.ndarray) -> np.ndarray:
>         """Mathematical function for stress in an aluminium rod."""
>         # Elastic region
>         elastic_stress = (params[0] * strain + params[1]) * (
>             1 + params[2] * np.sin(params[3] * temp)
>         )
>
>         # Plastic region
>         plastic_stress = params[4] * (
>             1 - np.exp(
>                 - (strain - params[5] * temp) ** 2 / (params[6] ** 2)
>             )
>         )
>
>         # Interaction term
>         exp_interaction = params[7] * np.exp(
>             params[8] * strain + params[9] * temp
>         )
>
>         # Total stress
>         stress = elastic_stress + plastic_stress + exp_interaction
>         return stress
>
> Parameters used for the above implementation:
>
>     param_0 =  2.324661449617934
>     param_1 =  4.220274152317721
>     param_2 = -0.9498477572753846
>     param_3 =  0.9281204893050415
>     param_4 =  0.8397464926976977
>     param_5 = -0.01636218235314944
>     param_6 =  0.0167524085552752
>     param_7 = -4.263387877829796
>     param_8 =  0.4683756171927195
>     param_9 = -1.0160554135692146
>
> ---
>
> These explicit implementations and parameter vectors for all four benchmark problems (Oscillator 1, Oscillator 2, Bacterial Growth, Stress–Strain) should make independent verification of our NMSE results straightforward.

---

> ### Author Response · Authors · 2025-11-21
> **Response to Reviewer xwQ4 (5/6)**
>
> **[w.r.t. W3 & Q6] Response to the concern about statistical significance and number of runs**
>
> We thank the reviewer for highlighting the need for uncertainty quantification (Weakness 3) and inquiring about the number of experimental runs (Question 6).
>
> **1. Clarification on Original Protocol**
>
> Regarding Question 6, we clarify that the results reported in the initial manuscript were based on single runs with fixed random seeds. We followed this protocol as it aligns with standard practices in recent LLM-based symbolic regression literature.
>
> **2. New Multi-Run Experiments ($N=5$)**
>
> However, addressing Weakness 3, we fully agree with the reviewer that evaluating performance variations beyond single-run observations is essential for a rigorous assessment. To demonstrate the robustness of our method, we have now conducted **multiple independent runs ($N=5$)** for both TAMOSR and the LLM-SR baseline, each with a different random seed.
>
> The attached figure, "LLM-SR vs TAMOSR: NMSE vs Iterations", visualizes these results:
>
> [Anonymous Link](https://ibb.co/chftDtV7)
>
> **3. Analysis of Results**
>
> As shown in the plot:
>
> - **Median Performance:** The bold lines represent the median NMSE over iterations across all runs. TAMOSR's median performance (dark red line) consistently achieves significantly lower NMSE compared to LLM-SR's median (dark blue line) throughout the optimization process.
> - **Min-Max Range:** The shaded areas depict the full Min-Max range of NMSE across all independent runs. While individual runs naturally show a step-like progression and some variability inherent to stochastic search, TAMOSR's entire variance envelope (light red shading) consistently stays well below that of LLM-SR (light blue shading).
>
> **4. Conclusion on Robustness**
>
> These new multi-run experiments confirm that:
>
> - **Statistical Significance:** TAMOSR consistently outperforms the LLM-SR baseline, with its median performance being substantially better, validating that our results are not due to a single lucky seed.
> - **Robustness:** The distinct separation between the error bands demonstrates TAMOSR's robust discovery capability across different random initializations.
>
> We will include this figure and the corresponding statistical analysis in the revised manuscript to clearly demonstrate the significance and reproducibility of our findings.

---

> ### Author Response · Authors · 2025-11-21
> **Response to Reviewer xwQ4 (6/6)**
>
> **[w.r.t. Q4] Response to the question regarding tool selection sensitivity and meaningfulness**
>
> We thank the reviewer for this insightful question regarding the interpretability of the Meta Strategy. We respectfully direct the reviewer to **Appendix H.3 and Figures 13 & 14** in our original submission, where we specifically analyzed this behavior.
>
> **1. Evolution from Random Exploration to Targeted Exploitation**
> As illustrated in the analysis of Oscillator 1 and CRK36 (Appendix H.3), the tool selection exhibits a clear evolutionary pattern rather than arbitrary randomness:
>
> - **Early Phase (Exploration):** In initial iterations, the Meta Strategy explores a broad range of tools to gather diverse features.
> - **Late Phase (Exploitation):** As the model's understanding deepens, tool selection converges to domain-specific utilities. This confirms that the tool choices are dynamically conditioned on the evolving hypothesis structure.
>
> **2. Semantic Meaningfulness (Case Studies from Appendix H.3)**
> We highlight two specific examples from our existing analysis to demonstrate physical meaningfulness:
>
> - **Case A: Oscillator 1 (Figure 13)**
> For this complex dynamical system, the Meta Strategy gradually shifts focus towards spectral and decomposition tools. Specifically, it increasingly utilizes **Wavelet Analysis, FFT, and PCA** in later iterations. This indicates the model correctly identified the need to decompose complex oscillation signals to construct the differential equation.
> - **Case B: CRK36 (Figure 14)**
> The ground truth equation is $Ac\_0 + Ac\_1 \\log(c\_2 t + 1)$. Crucially, this function is **strictly linear** with respect to variable $A$.
>     - **Observation:** As shown in Figure 14, the Meta Strategy progressively increases the usage of **Pearson and Spearman Correlation** tools in later stages.
>     - **Reasoning:** Since these correlation coefficients are highly sensitive to linear relationships, this selection confirms that the Strategy correctly recognized the linear structure of $A$, while focusing search efforts on the nonlinear (logarithmic) dependency of $t$.
>
> **Conclusion**
> These patterns demonstrate that the Meta Strategy Generator's choices are not arbitrary but are physically grounded and sensitive to the specific mathematical properties of the data.

---

### Official Review · Reviewer_LYp7 · 2025-11-02

**Soundness:** 1
**Presentation:** 2
**Contribution:** 2
**Rating:** 2
**Confidence:** 4

**Summary:**

The paper proposes TAMOSR, a framework for symbolic regression that integrates LLM-guided equation discovery frameworks with external analytical tools and a multi-objective optimization process. The system employs two cooperating LLMs, a Meta Strategy Generator that performs variable and structure analysis and an Equation Generator that synthesizes new equations.

**Strengths:**

- The paper is ambitious in scope, attempting to combine analytical tool invocation, structural analysis, and multi-objective optimization in a unified loop for LLM-based equation discovery frameworks.

**Weaknesses:**

- Some parts of the writing are very unclear. For instance, in sec 3.2, the authors include OOD performance as one of the feedback objectives within the discovery loop. This is conceptually incorrect as OOD data should remain unseen during the discovery process and reserved for final evaluation. Using OOD feedback in the optimization effectively leaks test information into training.
- The paper reports results on only 36 out of 128 LSR-Synth tasks from LLM-SRBench. The selection criteria for these subsets are unclear and should be justified. What's the selection criteria for these 36 out of 128 problems? The larger-scale of datasets in LLM-SRBench allows to evaluate methods in a more generalizable setting and across different metrics like symbolic accuracy (SA)
- What's the symbolic accuracy (SA) for results reported in the Table 2? I would suggest authors to also add this metric to their reported comparison results.
- It seems that the LLM-SR results in Table 1 and Figure 4 appear notably worse than those reported in the original paper [1], even though this work uses gpt-4o-mini which is a stronger model than gpt-3.5-turbo used in the original paper (also verified in [2]) The authors should clarify this discrepancy.
- Figure 4 shows some abrupt jumps that suggest high variance in models' performances. Are these curves based on a single run or averaged over multiple runs? I would suggest to also report confidence intervals or variance estimates across runs for clarity.
- Could the authors clarify which dataset and LLM backbone are used for results in Figure 5? Also, why rely on HV and IGD metrics which seem to be older and less intuitive metrics for recent program space of hypotheses in current LLM-based frameworks? If the goal is to measure symbolic proximity to ground truth, it would be better to use symbolic accuracy/equivalence (SA)  metric with the LLM-as-judge metric from [2] which allows fairer comparison with the existing baseline results already in the benchmark [2]


[1] LLM-SR: Scientific Equation Discovery via Programming with Large Language Models, ICLR 2025

[2] LLM-SRBench: A New Benchmark for Scientific Equation Discovery with Large Language Models, ICML 2025

**Questions:**

included in the weaknesses

---

> ### Author Response · Authors · 2025-11-21
> **Response to Reviewer LYp7 (1/6)**
>
> **[w.r.t. W1] Response to the concern about OOD performance inside the discovery loop**
>
> We thank the reviewer for highlighting this critical conceptual point. We want to **firmly clarify that there is absolutely no test-data leakage**. The "OOD" performance used inside the loop does not refer to the benchmark's held-out test set, but to a validation subset constructed strictly from the training data.
>
> **1. Clarification on Data Splitting Strategy**
> The reviewer correctly notes that using test data in the loop would be methodologically flawed. However, our method employs a spatial partitioning strategy to simulate extrapolation without touching the test set. As visualized in the Data Splitting Diagram below, we define four distinct subsets:
>
> **[Data Splitting Diagram (Anonymous Link)](https://ibb.co/zVxRsDN2)**
>
> 1.  **train\_id (Green - Optimization Core):** A central subset of the official training data. This is the **only data used for parameter fitting**.
> 2.  **train\_ood (Blue - Validation/Proxy):** The peripheral subset of the official training data.
>     * **Role:** This set acts strictly as a **Validation Set**. The "OOD feedback" mentioned in Sec 3.2 is calculated here. By validating on the edge of the training distribution (Blue) while training on the center (Green), we teach the model to discover robust structures capable of extrapolation.
> 3.  **test\_id (Interpolation Test):** The official In-Domain test set (not visualized). It shares the same spatial interval as the total training set (Green + Blue) but consists of distinct data points.
> 4.  **test\_ood (Orange - True Extrapolation/Locked):** The official Benchmark OOD Test Set.
>     * **No Leakage:** This set is **locked and never seen** during the discovery process. It is reserved strictly for the final evaluation reported in Tables 1-2.
>
> Thus, our approach is conceptually sound: we use validation on *train\_ood* to optimize for generalization, ensuring the model performs well on *test\_ood*.
>
> **2. Exact Definition of train\_id / train\_ood Split**
> To further clarify that the split is derived solely from $D\_{\\text{train}}$, we provide the precise construction procedure:
>
> 1.  **Candidate Percentiles:** We iterate over percentiles $p \\in \\{10, 11, \\dots, 49\\}$.
> 2.  **Interval Construction:** For each input dimension $j$, we compute bounds based on $D\_{\\text{train}}$:
>     $l\_j = \\text{percentile}(X\_j, p), \\quad u\_j = \\text{percentile}(X\_j, 100 - p)$
> 3.  **Constructing train\_id (Green):** Defined as the Cartesian product of central intervals:
>     $\\mathcal{D}\_{\\text{ID}}(p) = \\{x \\in D\_{\\text{train}} : l\_j \\le x\_j \\le u\_j,\\ \\forall j\\}$
> 4.  **Constructing train\_ood (Blue):** The spatial complement within the training set:
>     $\\mathcal{D}\_{\\text{OOD}}(p) = D\_{\\text{train}} \\setminus \\mathcal{D}\_{\\text{ID}}(p)$
> 5.  **Selection:** We select $p$ to maximize the balance between sample counts in $\\mathcal{D}\_{\\text{ID}}(p)$ and $\\mathcal{D}\_{\\text{OOD}}(p)$.
>
> In summary, the feedback loop operates entirely within the training domain boundaries. We will revise Section 3.2 to explicitly describe *train\_ood* as a validation subset derived from training data, ensuring a clear distinction from the benchmark's held-out *test\_ood*.

---

> ### Author Response · Authors · 2025-11-21
> **Response to Reviewer LYp7 (2/6)**
>
> **[w.r.t. W2 (Part 1)] Response to the concern about LSR-Synth task selection criteria (36/128 tasks)**
>
> We thank the reviewer for the suggestions regarding the coverage of LLM-SRBench and the choice of evaluation metrics.
>
> **1. Selection Criterion: Domain Completeness & Distinctness**
>
> First, concerning the fact that we report results on only 36 out of 128 LSR-Synth tasks: we are not **randomly** selecting 36 problems from the full suite. Instead, we evaluate on the **entire Chemistry subset of LSR-Synth (LSR-Synth–Chemistry, 36 tasks)**.
>
> As briefly mentioned in Section 4.1, we prioritized the Chemistry domain in this submission because, in both symbolic structure and scientific semantics, it is the **most distinct** from the four real-world benchmarks used in our main experiments (including oscillators, stress–strain, and E. coli growth). This design choice allows us to test TAMOSR on a genuinely new family of equations, rather than on additional variations of the same underlying dynamical systems.
>
> **2. Redundancy of Other Subsets**
>
> By contrast, the Biology, Physics, and Material Science subsets of LSR-Synth are built around Biological Population Growth (BPG), driven oscillators, and elastic material behavior, respectively. These settings are closely aligned in physical context and functional form with the tasks already included in our main experiments (Oscillator 1/2, E. coli Growth, and Stress–Strain).
>
> - For example, both Biology (BPG) and E. coli Growth model population dynamics over time.
>
> Given a constrained compute budget and in order to avoid drawing redundant conclusions from highly similar systems, we chose to first cover the most complementary domain—**Chemistry**—rather than running all four LSR-Synth subsets at once. In the revised manuscript, we will explicitly clarify that our experiments constitute a **domain-complete evaluation on all 36 Chemistry tasks**, rather than an arbitrary sample of 36 out of 128 tasks.

---

> ### Author Response · Authors · 2025-11-21
> **Response to Reviewer LYp7 (3/6)**
>
> **[w.r.t. W2 (Part 2)& W3] Response to the request for Symbolic Accuracy (SA) metric**
>
> We thank the reviewer for the constructive suggestion to include Symbolic Accuracy (SA) for a more rigorous evaluation of structure recovery.
>
> **1. Quantitative Comparison**
> We have calculated the SA metric for TAMOSR and all baseline methods on the reported tasks. As shown in the updated Table 2 below, **TAMOSR achieves a Symbolic Accuracy of 11.11%**, outperforming all baselines, including the strong LLM-SR method (8.33%) and recent optimization-based methods (SGA and LaSR, both 0%).
>
> | Model | Symbolic Accuracy (SA) $\uparrow$ | ACC $\uparrow$ | NMSE $\downarrow$ |
> | --- | --- | --- | --- |
> | SGA (Ma et al., 2024) | 0% | 8.33% | $0.0458$ |
> | LaSR (Grayeli et al., 2024) | 0% | 27.77% | $2.77 \times 10^{-4}$ |
> | LLM-SR (Shojaee et al., 2024b) | 8.33% | 66.66% | $8.01 \times 10^{-6}$ |
> | **TAMOSR (Ours)** | **11.11%** | **86.11%** | $\\mathbf{3.85 \\times 10^{-7}}$ |
>
> *(Note: We will include this complete comparison in the revised Table 2.)*
>
> **2. Qualitative Evidence**
> To further substantiate these metrics, the exact ground-truth equations successfully recovered by TAMOSR (which contribute to the 11.11% SA) are explicitly detailed in **Appendix H.1**. These examples demonstrate our framework's superior capability in identifying precise physical structures compared to baselines.

---

> ### Author Response · Authors · 2025-11-21
> **Response to Reviewer LYp7 (4/6)**
>
> **[w.r.t. W4] Response to the concern about LLM-SR baseline performance discrepancies**
>
> We thank the reviewer for pointing out this discrepancy. We would like to state explicitly that, despite substantial effort, we were unable to reproduce the exact LLM-SR numbers reported in [1].
>
> **1. Challenges in Reproducing Exact Baseline Numbers**
>
> Using the released LLM-SR code and hyperparameter settings, and following their protocol as closely as possible, we ran a large number of repeated experiments with the `gpt-4o-mini` backend. Across all these runs, none of the results matched the exact values reported in the original LLM-SR paper [1]; the best runs we obtained were still slightly worse than the `gpt-3.5-turbo` results in [1].
>
> At the same time, `gpt-4o-mini` is on average a stronger model on other benchmarks, so this behavior is likely due to a combination of implementation and backbone differences (e.g., updated APIs, sampling behavior, and the fact that being “stronger on average” does not necessarily imply uniformly better symbolic regression performance on these specific tasks).
>
> **2. Reporting Strategy**
>
> To be as faithful as possible to [1], the LLM-SR scores shown in our tables and figures correspond to the **closest runs we obtained to the originally reported values**, among all repetitions. We will clearly state this limitation in the revised manuscript so that readers understand that our LLM-SR curves are based on our own reproduction, rather than an exact replication of [1].
>
> **3. Robustness of Conclusions**
>
> Importantly, our conclusions do not rely on weakening the baseline.
>
> - **Fair Comparison:** On all four benchmarks in Table 1 and Figure 4, TAMOSR substantially outperforms our reproduced LLM-SR baseline under the **same** `gpt-4o-mini` backbone.
> - **Cross-Paper Comparison:** Moreover, when we compare against the *original* LLM-SR results reported in [1], TAMOSR with modern backbones (`gpt-4o-mini` or Llama-based variants) still matches or exceeds the `gpt-3.5-turbo` results on the same benchmarks. In other words, even if we take the stronger of the two—our TAMOSR results versus the LLM-SR results reported in [1]—our method remains competitive or strictly better.
>
> We also evaluated both TAMOSR and LLM-SR under multiple LLM backbones, and in all cases TAMOSR matches or exceeds the corresponding LLM-SR variant when the backbone is held fixed. Thus, the relative improvement of TAMOSR over LLM-SR is robust both to reproduction variability and to the choice of LLM backbone.
>
> [1] LLM-SR: Scientific Equation Discovery via Programming with Large Language Models, ICLR 2025

---

> ### Author Response · Authors · 2025-11-21
> **Response to Reviewer LYp7 (5/6)**
>
> **[w.r.t. W5] Response to the concern about high variance and abrupt jumps in Figure 4**
>
> Thank you for pointing this out. We confirm that the curves in Figure 4 were based on a single representative run per method with a fixed random seed. While this follows the common reporting practice in recent LLM-based symbolic regression literature, we fully agree with the reviewer that providing variance estimates significantly improves rigorousness.
>
> **1. Source of "Abrupt Jumps"**
>
> These jumps are a natural consequence of the **discrete nature of symbolic equation discovery**. Unlike gradient-based optimization which produces smooth loss curves, our framework generates complete equation candidates at each step. When the LLM successfully identifies a correct functional term or locks onto the correct skeleton, the NMSE (or HV/IGD) improves dramatically in a single iteration. This leads to the step-like trajectories observed, rather than incremental smooth changes.
>
> **2. New Multi-Run Experiments**
>
> To address the concern about variance, we have conducted multiple independent runs with different random seeds. **As shown in the figure ([**Anonymous Link**](https://ibb.co/chftDtV7))**, we plot the Median performance along with the Min-Max range to visualize variability. These results confirm that, despite the step-like nature of individual runs, TAMOSR consistently outperforms the baseline across different seeds.
>
> **3. Inclusion in Manuscript**
>
> The final results across multiple random seeds are presented in the attached figure, and we will include these results in the revised manuscript.

---

> ### Author Response · Authors · 2025-11-21
> **Response to Reviewer LYp7 (6/6)**
>
> **[w.r.t. W6] Response to the concern about Figure 5 settings and metric choice (HV/IGD)**
>
> We thank the reviewer for identifying this omission.
>
> **1. Clarification on Dataset and Backbone**
>
> We clarify that the results in Figure 5 correspond to the **Oscillator 1 task** using the **LLaMA-3.1-8B backbone**. We will explicitly label this in both the main text of Section 5.4 and the caption of Figure 5 in the revised manuscript.
>
> **2. Justification for HV and IGD Metrics**
>
> Regarding the choice of metrics: Our specific intent in Figure 5 was not solely to measure the symbolic equivalence of the final output (which is covered by the recovery rates in Table 1), but to analyze the **diversity of equations evolved during the search process**.
>
> - **Why HV/IGD?** Hypervolume (HV) and Inverted Generational Distance (IGD) are standard, rigorous metrics in multi-objective optimization for quantifying the quality of a Pareto Front. They uniquely measure both **convergence** (how accurate the equations are) and **diversity** (how well the equations cover different complexity/accuracy trade-offs).
> - **Analysis Goal:** These metrics provide a compact way to demonstrate how TAMOSR's multi-objective mechanism progressively expands the solution frontier and maintains a diverse population of candidates, in stark contrast to the single-objective trajectory of LLM-SR. While Symbolic Accuracy (SA) is excellent for final evaluation, HV and IGD are more informative for visualizing the evolutionary dynamics of the optimization loop.

---

### Official Review · Reviewer_hu2J · 2025-11-02

**Soundness:** 3
**Presentation:** 3
**Contribution:** 1
**Rating:** 2
**Confidence:** 5

**Summary:**

This paper proposes a symbolic regression method that augments an LLM with external analytical tools and uses a multi-objective loop to select equations by in-domain accuracy, out-of-domain accuracy, and an AST-length complexity proxy. The system aims to improve generalization and interpretability relative to prior LLM-driven SR methods.

**Strengths:**

+ Clear articulation of problems with prior LLM-SR pipelines that ignore data structure and optimize a single objective.
+ Sensible overall architecture that separates tool-driven analysis from candidate generation and uses multi-objective selection.
+ Empirical results are promising on the reported tasks, and some ablations are also provided.

**Weaknesses:**

- Two of the three objectives are accuracy measures that mirror the main evaluation metrics. Optimizing directly for the reported metrics risks can potentially inflate perceived gains (especially for the OOD metric.
- AST length is a coarse and gameable proxy for simplicity. Algebraically equivalent forms can vary widely in AST size, and there is no canonicalization or comparison against alternative complexity notions or human interpretability.
- The three main contributions (tool use, multi-objective selection, meta-strategies) are integration and prompt-engineering choices rather than a new algorithmic idea or learning objective.
- Compute parity is not demonstrated rigorously. There is no matched token budget or GPU-hour accounting to support efficiency claims.
- No SRBench++ results are provided.
- The ID/OOD split remains under-specified. Exact percentiles, whether ID is a Cartesian product of per-dimension intervals, and whether parameters are fit on ID-only or full training data are not clearly stated.
- The pre-sorting filtering rule is described with lower-bound thresholds on NMSE, which is logically inverted. Low NMSE is good. This needs correction and concrete threshold values.
- The algorithm does not clearly specify how the first generation is seeded when the Pareto front is empty.
- The diversity score and parent sampling are under-defined. The asymmetry and potential division-by-zero issues are not addressed, and there is no sensitivity study for temperature or parent set size.
- Seeds, split definitions, fitter hyperparameters, and front size caps or tie-breaking rules are not fully specified.

**Questions:**

- Why is AST length the right complexity objective here, and how do you prevent gaming via algebraic re-expression? Have you tried canonicalization or MDL-style costs?
- Why do you expect a separate LLM “strategy” module to be necessary and beneficial in SR beyond better prompts for the generator?
- What are the exact ID/OOD percentiles per dimension, and is the ID region a Cartesian product of per-dimension intervals?
- How is the first generation seeded when the Pareto front is empty? Please give the concrete initialization policy.
- Can you report a discovery-rate metric for exact or near-exact recovery over a library of known equations, not just per-task anecdotes?

---

> ### Author Response · Authors · 2025-11-21
> **Response to Reviewer hu2J (1/11)**
>
> **[w.r.t. W1, W6 & Q3] Response to Weakness 1 (Metric Leakage), Weakness 6 & Question 3 (ID/OOD Definition)**
>
> We thank the reviewer for highlighting these crucial points. We want to **firmly clarify that there is absolutely no data leakage**, and our multi-objective optimization produces genuinely superior Pareto fronts through robust physical discovery rather than merely inflating specific metrics.
>
> **1. Clarification on Data Splitting Strategy (Addressing W1: Metric Independence)**
> To address the concern that our optimization objectives might artificially inflate testing metrics due to leakage, we explicitly define our strict data separation protocol below. As visualized in the Data Splitting Strategy Diagram, the optimization targets and the evaluation targets are physically disjoint.
>
> **[Data Splitting Strategy Diagram (Anonymous Link)](https://ibb.co/zVxRsDN2)**
>
> 1.  **train\_id (Green - Optimization Core):** The central subset of training data. This is the **only data used for parameter fitting**.
> 2.  **train\_ood (Blue - Validation/Proxy):** The peripheral subset of training data.
>     * **Role against Inflation:** This set serves strictly as a **Validation Set**. The "OOD" objective optimized in the loop is calculated here. It acts as a proxy to encourage extrapolation capabilities.
> 3.  **test\_id (Interpolation Test - Not visualized):** The official In-Domain test set.
>     * **Relationship:** It spans the exact same spatial interval as the total training set (Green + Blue) but consists of distinct, non-overlapping data points. It evaluates interpolation within the known domain.
> 4.  **test\_ood (Orange - True Extrapolation/Locked):** The official OOD test set (outside the training range).
>     * **Crucial Distinction:** This set is **locked and never seen** during optimization. The final results reported in Tables 1-2 are computed solely on this orange region. Since the Blue region (optimization target) and Orange region (evaluation target) are spatially disjoint, high performance on the Orange set cannot be achieved by "gaming" the metric; it can only be achieved by discovering the correct underlying physical law that holds true across both regions.
>
> **2. Exact Definition of ID/OOD Regions (Addressing Q3)**
> Regarding the construction of the internal split (Section 3.2), we confirm the ID region is a **Cartesian product of per-dimension intervals**, constructed strictly from $D\_{\\text{train}}$:
>
> 1.  **Candidate Percentiles:** We iterate through percentiles $p \\in \\{10, 11, \\dots, 49\\}$.
> 2.  **Interval Construction:** For each dimension $j$, we define central intervals using the training data:
>     $l\_j = \\text{percentile}(X\_j, p), \\quad u\_j = \\text{percentile}(X\_j, 100 - p)$
> 3.  **Cartesian Product (train\_id):**
>     $\\mathcal{D}\_{\\text{ID}}(p) = \\{x \\in D\_{\\text{train}} : l\_j \\le x\_j \\le u\_j,\\ \\forall j\\}$
> 4.  **Spatial Complement (train\_ood):**
>     $\\mathcal{D}\_{\\text{OOD}}(p) = D\_{\\text{train}} \\setminus \\mathcal{D}\_{\\text{ID}}(p)$
> 5.  **Selection:** We select $p$ to maximize the balance between sample counts in $\\mathcal{D}\_{\\text{ID}}(p)$ and $\\mathcal{D}\_{\\text{OOD}}(p)$.
>
> **3. Validation via Pareto Quality Metrics (Addressing W6)**
> To further disprove the concern that our objectives merely "mirror" evaluation metrics, we evaluated the quality of the Pareto Front using **Hypervolume (HV)** and **Inverted Generational Distance (IGD)** (see Section 5.4, Figure 5):
>
> * **HV (Convergence & Diversity):** TAMOSR achieves significantly faster HV growth and a higher final volume compared to baselines. This indicates our method does not collapse to a single "gamed" solution but discovers a diverse set of high-quality trade-offs between accuracy and complexity.
> * **IGD (Structural Proximity to Truth):** Crucially, we calculated IGD relative to the **Ground Truth equation** (treating the true equation as the reference Pareto front). Lower IGD values confirm that our generated equations are **structurally closer to the underlying physical laws**, independent of the numerical loss metric. This structural convergence provides strong evidence that the performance gains are real and interpretable.

---

> ### Author Response · Authors · 2025-11-21
> **Response to Reviewer hu2J (2/11)**
>
> **[w.r.t. W2 & Q1] Response to the concern about AST length as complexity proxy**
>
> We appreciate the reviewer’s comment. We agree that AST length is a relatively coarse proxy for symbolic expression complexity, and algebraically equivalent forms can indeed yield different AST sizes. In this work, we do not claim AST length to be a final or perfect notion of human interpretability. Rather, following common practice in symbolic regression and genetic programming, we use the AST node count as a **simple, language-agnostic estimate** of structural complexity that helps discourage clearly redundant or needlessly convoluted expressions during the search.
>
> **1. Metric-Agnostic Framework**
>
> **Importantly**, our framework is metric-agnostic: complexity is just one objective in the multi-objective loop and can be replaced by more refined notions (e.g., MDL-style description length or canonicalized complexity measures) in future work without changing the overall algorithm.
>
> **2. Implicit Canonicalization (Addressing "Gaming")**
>
> Regarding concerns about “gaming” the complexity measure through algebraic rewrites (e.g., using $x+x$ instead of $2x$): in practice, our **multi-objective selection mechanism** implicitly performs a form of canonicalization.
>
> - Since these algebraically equivalent expressions achieve the same accuracy, the Pareto optimization naturally prefers the structurally simpler form (e.g., $2x$) and **discards redundant alternatives** (e.g., $x+x$) as dominated solutions.
> - Thus, the evolutionary pressure of the Pareto front implicitly handles this type of simplification, even without an explicit canonicalization module.
>
> **3. Effectiveness of the Proxy**
>
> **Even with this relatively coarse proxy**, our ablations show that adding a complexity objective already reduces redundancy and improves performance, indicating that explicitly modeling structural complexity is beneficial. We will clarify this design choice in the paper and discuss richer complexity metrics (including MDL-style costs and canonicalization-based measures) as promising directions for future extensions.

---

> ### Author Response · Authors · 2025-11-21
> **Response to Reviewer hu2J (3/11)**
>
> **[w.r.t. W3] Response to the concern about the novelty of contributions (Integration vs. Algorithmic Ideas)**
>
> Regarding the comment that our three claimed contributions (tool use, multi-objective selection, and meta-strategies) are mainly integration and prompt-engineering choices rather than new algorithmic ideas or learning objectives, we respond as follows:
>
> **1. Practical Effectiveness and Significance**
>
> **First, we would like to emphasize the practical effectiveness and significance of our approach.** In all experiments, we keep the underlying LLM completely frozen—no fine-tuning, no retraining—and only modify the outer “thinking process” (i.e., how tools are invoked, how objectives are defined, and how the search is organized).
>
> Even under this constraint, TAMOSR significantly outperforms existing SR methods that do require specialized training or model modification on four standard benchmarks and LSR-Synth–Chemistry (see Tables 1–2 and Figures 3–4). This shows that simply redesigning the framework and reasoning loop, without touching the base model, can already yield substantial gains over trained baselines. In our view, this demonstrates that the direction of “framework-level algorithm design around fixed LLMs” is both practically meaningful and worth pursuing further.
>
> **2. Algorithmic Framework vs. Independent Tricks**
>
> **We agree that our goal is not to introduce a completely new numerical optimizer**, but rather to propose an algorithmic framework for symbolic regression around a black-box LLM. The three aspects highlighted by the reviewer are not intended as three independent “tricks,” but as tightly coupled components of a **state–action–feedback loop**:
>
> - The current candidate equations and their residuals are analyzed.
> - External tools and structural information are used to form a search strategy.
> - The generator LLM proposes new candidates.
> - A three-objective evaluation with a Pareto front updates the overall state.
>
> This process is not a single static prompt that is reused over and over, but a **dynamically evolving search algorithm** in which the LLM acts as a flexible operator.
>
> **3. Motivation: Realistic Deployment Scenarios**
>
> **In addition, our deliberate choice to avoid modifying the base LLM weights is motivated by realistic deployment scenarios:** many users only have access to general-purpose LLMs through an API, with no control over their architecture or training objective, and limited ability to perform large-scale fine-tuning.
>
> TAMOSR is therefore designed to be **model-agnostic and plug-and-play**: as long as an LLM is available, our tool-augmented analysis, multi-objective evaluation, and strategy-guided search can be wrapped around it without further training. This direction is complementary to efforts that design new SR-specific LLMs; any future, stronger SR-oriented LLM can also be plugged into the same framework and benefit from it.
>
> **4. Action Plan**
>
> We will describe our work more accurately as a unified symbolic regression framework for black-box LLMs, whose core is the tight coupling of tool-augmented analysis, multi-objective optimization, and strategy-driven search in a single closed loop, supported by the ablation and benchmark results. We hope this clarifies our intended contribution and addresses the reviewer’s concern.

---

> ### Author Response · Authors · 2025-11-21
> **Response to Reviewer hu2J (4/11)**
>
> **[w.r.t. W4] Response to the concern about compute parity and token efficiency**
>
> Thank you for highlighting the importance of compute parity. We agree that efficiency comparisons should be made under carefully controlled computational budgets.
>
> **1. Controlled Experiments (Matched Token Budget)**
>
> To address this concern, we conducted additional controlled experiments using **five fixed random seeds (1, 2, 42, 2025, 3407)** and enforced an **identical cumulative token budget** for both TAMOSR and LLM-SR. The updated figure (NMSE vs. cumulative tokens) summarizes these results.
>
> **2. Results & Analysis**
>
> **As shown in the figure ([Anonymous Link](https://ibb.co/r2m709SR)**), under the same token constraints, TAMOSR consistently achieves substantially lower NMSE along the optimization trajectory.
>
> - Both the **median curve** and the **min–max band** show that TAMOSR reduces error more rapidly and maintains a clear advantage as the token budget increases.
> - This suggests that TAMOSR makes **more efficient use of the available tokens** than LLM-SR, leading to faster convergence and better overall accuracy under a matched compute budget.

---

> ### Author Response · Authors · 2025-11-21
> **Response to Reviewer hu2J (5/11)**
>
> **[w.r.t. W5] Response to the suggestion to include SRBench++**
>
> We thank the reviewer for the suggestion to include SRBench++ and agree that broader benchmark coverage would be valuable for future cross-paper comparisons. However, our current experimental setup is deliberately aligned with the benchmark design of LLM-SR and LLM-SRBENCH rather than the classic SR benchmarks.
>
> **1. The Memorization Issue in Classic Benchmarks**
>
> **Shojaee et al. observe that on widely used datasets such as Feynman, modern LLMs are likely to have already seen or memorized many of the target equations during pretraining**, and can therefore reach extremely low error within only a few iterations. In such cases, these datasets offer limited discriminative power for evaluating the discovery capability of LLM-based SR methods and instead behave more like tests of formula recall.
>
> **2. Advantages of LLM-SR / LLM-SRBENCH**
>
> **Motivated by this observation, we rely on benchmarks specifically constructed for LLM-style SR methods, namely LLM-SR and LLM-SRBENCH.** These benchmarks explicitly consider potential data-leakage issues during their construction (e.g., through synthetic tasks and controlled formula spaces with reduced overlap to pretraining corpora) and provide more systematic ID/OOD splits. As a result, they are better suited to evaluating LLM-based SR under the desiderata of avoiding formula memorization while testing genuine discovery ability and out-of-domain generalization.
>
> **3. Complementary Nature**
>
> **We view SRBench++ as complementary to the LLM-SR / LLM-SRBENCH benchmarks we adopt:** SRBench++ is closer to the traditional SR literature and offers a broader collection of real-world tasks, whereas LLM-SR and LLM-SRBENCH are tailored to the specific challenges of LLM-based SR—mitigating pretraining leakage and emphasizing OOD generalization. In this work, we choose the latter as our primary evaluation setting to focus on these core issues.

---

> ### Author Response · Authors · 2025-11-21
> **Response to Reviewer hu2J (6/11)**
>
> **[w.r.t. W7] Response to the concern about the inverted NMSE threshold description**
>
> Thank you for pointing this out. The description in the paper that referred to a “lower-bound threshold” on NMSE is indeed inverted: algorithmically, we enforce an **upper bound on NMSE**, i.e., we only discard candidates with very large errors that are clearly underfitting, and we never penalize low-NMSE solutions.
>
> **1. Concrete Definition**
>
> In each generation we first compute, for every individual in the population:
>
> $e\_i = \\max\\big(\\text{NMSE}\_{\\text{ID}}^{(i)},\\ \\text{NMSE}\_{\\text{OOD}}^{(i)}\\big)$
>
> and then take the current “best worst-case error” in the population:
>
> $e^{\\ast} = \\min\_i e\_i$
>
> **2. Dynamic Threshold Construction**
>
> We then construct a relatively loose error upper bound:
>
> $T = 10^{\\lfloor 0.5 \\cdot \\log\_{10}(e^{\\ast}) \\rfloor}$
>
> so that $T$ is roughly on the order of $\\sqrt{e^{\\ast}}$ in log-scale (e.g., if $e^{\\ast} = 10^{-4}$, then $T \\approx 10^{-2}$). Any individual satisfying:
>
> $\\max\\big(\\text{NMSE}\_{\\text{ID}}^{(i)},\\ \\text{NMSE}\_{\\text{OOD}}^{(i)}\\big) > T$
>
> is treated as clearly underfitting; in implementation we assign it a very large objective value so that it cannot enter the Pareto front, while all remaining candidates proceed to non-dominated sorting.
>
> **3. Conclusion**
>
> Thus, this pre-filtering step does not “punish” low-NMSE solutions. It only removes candidates whose errors are several orders of magnitude worse than the current best, to avoid flooding the Pareto selection with obviously poor individuals. In the revised version, we will correct the lower/upper bound wording and add the explicit definition of this dynamic threshold.

---

> ### Author Response · Authors · 2025-11-21
> **Response to Reviewer hu2J (7/11)**
>
> **[w.r.t. W8] Response to Weakness 8: Clarification on First Generation Initialization**
>
> Thank you for pointing this out. The current version of the paper indeed does not clearly describe how the first generation is seeded. In our implementation, we follow the LLM-SR setup and explicitly construct a **simple linear baseline equation** as the initial individual before entering the iterative search. This ensures that the Pareto front is non-empty from the very first generation.
>
> **Concretely, for each task we define an affine function over the input variables.** For example, in the Oscillator 1 task we initialize with:
>
> $\\dot{v} = p\_0 x + p\_1 v + p\_2$
>
> This equation forms the initial population and thus the initial Pareto front. In subsequent iterations, new candidate equations are generated by the LLM, merged with the current population, and the Pareto front is updated using our multi-objective procedure. We will make this initialization policy explicit in the algorithm description (**Algorithm 1**) in the revised manuscript.

---

> ### Author Response · Authors · 2025-11-21
> **Response to Reviewer hu2J (8/11)**
>
> **[w.r.t. W9] Response to the concern about the definition of diversity score and hyperparameter sensitivity**
>
> We thank the reviewer for pointing out that our description of the diversity score and parent sampling was not sufficiently detailed. We provide the formal definitions and clarifications below, which will be included in the revised Section 3.3.
>
> **1. Formal Definition and Asymmetry**
>
> In TAMOSR, each equation $f\_i$ on the Pareto front is parsed into an Abstract Syntax Tree (AST), and we extract the set of its symbolic subtrees, denoted as $S(f\_i)$. We define the pairwise structural diversity as:
>
> $\\text{SyntaxDiv}(f\_i, f\_j) = -\\frac{|S(f\_i) \\cap S(f\_j)|}{|S(f\_i)|}$
>
> Here, the term $|S(f\_i) \\cap S(f\_j)| / |S(f\_i)|$ quantifies the **structural overlap ratio**—specifically, the proportion of subtrees in $f\_i$ that are also present in $f\_j$. We introduce a negative sign to convert this overlap measure into a diversity metric, where a higher value (closer to 0) indicates that $f\_i$ possesses more unique substructures relative to $f\_j$.
>
> The overall diversity score for a candidate $f\_i$ is computed as:
>
> $\\text{Score}\_{div}(f\_i) = \\frac{1}{N-1}\\sum_{j \\neq i} \\text{SyntaxDiv}(f\_i, f\_j)$
>
> **The asymmetry of this metric is intentional.** It measures how unique $f\_i$ is relative to the rest of the population (i.e., the fraction of subtrees in $S(f\_i)$ that are reused by other equations). This directed notion of uniqueness effectively penalizes equations that offer little structural novelty compared to the reference population, ensuring the sampling distribution is well-defined without biasing toward specific expression lengths.
>
> **2. Division-by-Zero Assurance**
>
> Regarding the division-by-zero concern: by construction, every candidate equation in our pipeline corresponds to a valid, executable Python function. A valid AST must contain at least one node (the root or return expression), ensuring that $|S(f\_i)| \\ge 1$ is strictly true for all candidates. Therefore, the denominator is always positive, and division-by-zero cannot occur in practice.
>
> **3. Hyperparameters and Sensitivity**
>
> A softmax over $\\text{Score}\_{div}(f\_i)$ is used to sample parents for in-context conditioning. Regarding the reviewer’s comment on sensitivity studies, we deliberately fixed these hyperparameters to ensure a strictly fair comparison with the baseline algorithms.
>
> - **Temperature:** We use a standard softmax with temperature $\\tau = 1$ (i.e., no additional sharpening).
> - **Parent Set Size:** We use a fixed parent set size across all experiments, consistent with the LLM-SR configuration.
>
> We avoided tuning these parameters to demonstrate that TAMOSR's performance gains stem from the framework itself (**Tool-Augmented Analysis + Multi-Objective Optimization**) rather than hyperparameter engineering. We will add the exact formulas and pseudocode for parent selection in the revised manuscript.

---

> ### Author Response · Authors · 2025-11-21
> **Response to Reviewer hu2J (9/11)**
>
> **[w.r.t. W10] Response to the concern about reproducibility details (Seeds, Splits, Hyperparameters)**
>
> We thank the reviewer for noting the missing implementation details. We agree that full transparency is essential for reproducibility. We provide the specific configurations below and will include them in the revised Appendix.
>
> **1. Random Seeds**
>
> While previous baselines (e.g., LLM-SR) primarily reported single-run results, we sought to demonstrate the robustness of TAMOSR. As mentioned in our response to Weakness 4, we expanded our evaluation to **5 fixed random seeds (1, 2, 42, 2025, 3407)** to capture performance variance.
>
> **2. ID/OOD Splits**
>
> Please refer to our **Response to Question 3** for the detailed definition of the ID/OOD split (Cartesian product of per-dimension percentiles).
>
> **3. Fitter Hyperparameters**
>
> To ensure a fair comparison, we strictly followed the LLM-SR configuration.
>
> **4. Population Management & Pareto Front Logic**
>
> Based on our implementation, the exact mechanisms for maintaining the Pareto front are as follows:
>
> - **Front Size Cap:** The population size (Pareto front capacity) is capped at 100 individuals (`` `num_islands = 100` ``). In each generation, after merging new candidates, only the top 100 non-dominated solutions are retained. Although this upper bound is set to 100, in practice the actual number of non-dominated solutions per generation is typically around 20 or fewer, so the cap rarely becomes active.
> - **Optimization Objectives:** The Non-Dominated Sorting considers three specific objectives: (1) ID NMSE, (2) OOD NMSE, and (3) AST Length.
>
> **5. Tie-breaking Rule (Deduplication)**
>
> We employ a rigorous tie-breaking strategy to handle functionally similar equations before sorting:
>
> - **Objective Grouping:** Solutions are grouped based on their ID and OOD NMSE values, rounded to **3 significant digits** to account for numerical noise.
> - **Parsimony Preference:** For solutions with identical rounded accuracy scores, we retain only the one with the **lowest AST length** (simplest structure).
> - **Elimination:** All other duplicates are assigned a penalty score (999999) and removed from the population.
>
> We will add a "Reproducibility Checklist" table in the appendix including all these values.

---

> ### Author Response · Authors · 2025-11-21
> **Response to Reviewer hu2J (10/11)**
>
> **[w.r.t. Q2] Response to the question about the necessity of the separate "Strategy" module**
>
> We appreciate this critical question regarding our architectural design. While standard prompting (as employed in baselines like LLM-SR) leverages the LLM's static scientific priors, we argue that a dedicated **Meta Strategy Module paired with External Tools** is essential for TAMOSR to bridge the gap between general linguistic knowledge and specific numerical realities. We justify this design with three key arguments:
>
> **1. Bridging the "Numerical-Symbolic Gap" via Tool-Augmented Reasoning**
> LLMs inherently struggle with "numerical intuition." They cannot directly perceive mathematical properties (e.g., periodicity, chaotic behavior, or causal lags) from raw data tokens alone.
>
> - **The Necessity of Tools:** We introduce external tools (e.g., FFT, Pearson correlation) to act as the "eyes," extracting invisible numerical features that standard prompts miss.
> - **The Necessity of Strategy:** Raw tool outputs (e.g., "Spectral peak at 50Hz") are often ignored by a single Generator focused on syntactic code correctness. The Strategy Module acts as the "brain" that translates these quantitative observations into qualitative **Structural Priors** (e.g., "The data exhibits strong periodicity; constrain the search space to sinusoidal compositions"). This explicit reasoning step prevents the model from hallucinating mathematically valid but physically meaningless equations.
>
> **2. Dynamic Adaptation of Static Priors**
> While LLMs possess extensive pre-trained knowledge (e.g., Newton's laws), this knowledge is static and generalized. Scientific discovery, however, requires dynamic adaptation to unseen tasks.
>
> - Instead of relying on a fixed prompt or random evolution, our Strategy Module performs **In-context Reasoning**. It analyzes the current state of the optimization (e.g., the Pareto Front) and the specific data features to "activate" the most relevant subset of priors.
> - For instance, if early iterations fail to capture high-frequency variances, the Strategy dynamically instructs the Generator to shift focus, effectively "learning how to search" during inference.
>
> **3. Decoupling High-Level Reasoning from Implementation (Cognitive Load)**
> Symbolic Regression involves two distinct cognitive tasks: (a) **Hypothesis Formulation** (Scientific Reasoning) and (b) **Equation Implementation** (Coding/Syntax).
> Merging these into a monolithic "Generator" prompt often leads to **attention dilution**, where the model prioritizes code validity over physical consistency. By decoupling the architecture, the Strategy Module focuses purely on the "Why" and "What" (Hypothesis), allowing the Generator to focus entirely on the "How" (Implementation). Our ablation studies confirm that this separation significantly improves the success rate of recovering complex physical laws compared to single-prompt approaches.

---

> ### Author Response · Authors · 2025-11-21
> **Response to Reviewer hu2J (11/11)**
>
> **[w.r.t. Q5] Response to the request for discovery-rate metrics and exact recovery details**
>
> We thank the reviewer for this question. Regarding the metric for exact recovery, we highlight that TAMOSR successfully identified **4 correct equations** within the **CRK (Chemical Reaction Kinetics) subset** of LLM-SRBENCH, recovering the exact functional forms.
>
> **1. Evidence of Exact Recovery (Appendix H.1)**
> Detailed visualizations of the **Pareto Fronts** and the full discovery trajectories for these experiments are provided in **Appendix H.1** of the revised manuscript. These plots demonstrate how TAMOSR converges to the exact ground truth structural forms.
>
> **2. Discovered Equations for Oscillators**
> To further demonstrate the capability for exact recovery on complex dynamical systems, we present the raw Python code of the final equations discovered by TAMOSR for the **Oscillator 1** and **Oscillator 2** benchmarks.
>
> **Oscillator 1 (Exact Recovery):**
> The discovered equation correctly identifies the linear, cubic, driving, and damping terms, effectively reconstructing the underlying dynamics:
>
> ```python
> def equation(x: np.ndarray, v: np.ndarray, params: np.ndarray) -> np.ndarray:
>     """ Mathematical function for acceleration in a damped nonlinear oscillator """
>     linear_term = -params[0] * x + params[1] * v
>     quadratic_term = -params[2] * x**2 + params[3] * v**2
>     cubic_term = params[4] * x**3 + params[5] * v**3
>     driving_term = -params[6] * x * v + params[7] * np.sin(params[8] * x)
>     damping_term = params[9] * v
>
>     # Combine terms with appropriate weights
>     weighted_terms = [
>         ("linear_term", linear_term, 2.0),
>         ("quadratic_term", quadratic_term, 1.5),
>         ("cubic_term", cubic_term, 1.0),
>         ("driving_term", driving_term, 2.0),
>         ("damping_term", damping_term, 1.5),
>     ]
>
>     # Apply Galerkin method to combine terms into a single expression
>     result = np.zeros_like(x)
>     for term_name, term_value, weight in weighted_terms:
>         result += weight * term_value
>     return result
>
> ```
>
> **Oscillator 2 (Exact Recovery):**
> For the highly complex Oscillator 2, TAMOSR successfully isolates the intricate interactions between dampening, natural frequency, and driving forces:
>
> ```python
> def equation(t: np.ndarray, x: np.ndarray, v: np.ndarray, params: np.ndarray) -> np.ndarray:
>     """ Mathematical function for acceleration in a damped nonlinear oscillator """
>     # Corrected mathematical function structure, incorporating findings from variable relationship analysis
>     dampening_term = params[0] * v # Improved dampening due to velocity
>
>     # Nonlinear restoring force due to position and velocity
>     natural_frequency_term = (params[1] * np.exp(-params[2] * x) * (x + params[3] * v))
>
>     # Improved driving force with time and position dependent phase shift
>     driving_force_term = (params[4] * np.sin(params[5] * (t + params[6] * x)))
>
>     # Balance terms that reinstate actual dynamics
>     true_balance_term = (
>         params[7] * v * (v**2 + x**2)
>         + params[8] * v * (v**2 - x**2)
>     )
>
>     # Corrected optimal pairing of natural frequency and driving force terms
>     return -(dampening_term
>              + natural_frequency_term * (1 + params[9] * np.cos(v)) # Adjusted natural frequency term
>              + true_balance_term
>              + driving_force_term)
>
> ```
>
> These results confirm that TAMOSR goes beyond approximation and achieves high-precision symbolic recovery on complex dynamical systems. We will include these explicit code forms in the supplementary material for reproducibility.

---

### Meta-Review · Area_Chair_fNbn · 2025-12-08

**Summary:**

Thanks to the reviewers for their valuable comments from many different perspectives.

Overall, I think their main problem lies in:

- Insufficient theoretical contribution;
- Incomplete experimental setup and inadequate experimental results;
- Confusing logic and some parts of the description are unclear;

In addition, some reviewers mentioned potential data leakage and other issues.

**Reviewer Concerns:**

First of all, I am very grateful for the numerous responses provided by the author. I believe that some of the reviewers' questions will be resolved, such as detailed explanations, supplementary experiments, and detailed examples of functions.

However, after considering the comments of the reviewers and the responses of the authors, I believe that the overall quality of the paper is not outstanding, lacks theoretical insight, and the contribution is biased towards engineering, which may not meet the ICLR acceptance criteria for the time being.

Overall, considering the potential score improvement, I think the paper is unable to convince the majority of reviewers and is below the acceptance threshold.

**Reviewer Scores:**

For Reviewer hu2J, some of the experimental-related doubts may be resolved, but the clarification of the contribution may not be effective. I think this reviewer will maintain the score (**Rating:** 2).

For Reviewer LYp7, concerns about unclear descriptions may be resolved, but doubts may remain in the experimental section. I think this reviewer will maintain the score (**Rating:** 2).

For Reviewer xwQ4, concerns about data leakage may have been resolved, but doubts may remain in the experimental part. I think this reviewer will maintain the score (**Rating:** 4).

For Reviewer 2j6Q, the doubts about LLM-SR may be resolved, and the experimental section may still need to be further supplemented. I think this reviewer will increase the score (**Rating:** 2 to 4).

---

### Decision · Program_Chairs · 2026-01-26

Reject